# REGRESSION LANGUAGE MODELS FOR CODE

## ABSTRACT

We study **code-to-metric regression**: predicting numeric outcomes of code executions, a challenging task due to the open-ended nature of programming languages. While prior methods have resorted to heavy and domain-specific feature engineering, we show that a single unified Regression Language Model (RLM) can simultaneously predict directly from text, (i) the memory footprint of code across multiple high-level languages such as Python and C++, (ii) the latency of Triton GPU kernels, and (iii) the accuracy and speed of trained neural networks represented in ONNX. In particular, a relatively small 300M parameter RLM initialized from T5Gemma, obtains >0.9 Spearman-rank on competitive programming submissions from APPS, and a single unified model achieves >0.5 average Spearman-rank across 17 separate languages from CodeNet. Furthermore, the RLM can obtain the highest average Kendall-Tau of 0.46 on five classic NAS design spaces previously dominated by graph neural networks, and simultaneously predict architecture latencies on numerous hardware platforms.

## 1 INTRODUCTION

Predicting metric outcomes from programs and source code is a valuable capability that has been intensely studied over the past few years, with varying names such as *performance prediction* and *static analysis*. The goal is to predict a useful metric, such as performance or efficiency, produced by executing a computation graph represented as either a high-level language such as Python, or low-level program such as XLA. Achieving high precision predictions would naturally lead to more informed decision-making and better optimizations of all aspects in computing, including systems design, hardware manufacturing, and scientific discovery. However, one of the fundamental challenges in this domain is feature engineering, that is, learning highly accurate regression models over data from highly non-tabular, graph-based representations, which ideally should also be transferrable and reusable for new tasks.

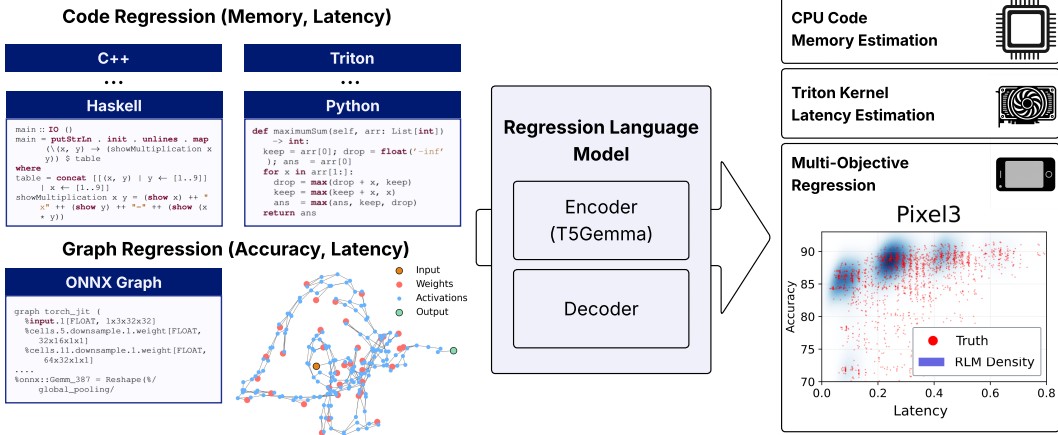

Figure 1: A Regression Language Model (RLM) is able to simultaneously read code from many different languages and compilation levels, and predict metrics such as accuracy, memory, and latency.

Recent work (Song et al., 2024) have proposed a promising yet simple regression method, "text-to-text regression", based on small customized language models which can be trained over large amounts of $(x, y)$ regression data represented as text. These *Regression Language Models* (RLMs) have shown promise over a variety of domains such as hyperparameter optimization (Song et al., 2024) and industrial systems (Akhauri et al., 2025), but up until now, it has been unknown whether such techniques can also be used for predictions over programs common to compilers and machine learning architectures. Below, we list our findings:

- A single, unified RLM initialized with a pretrained T5Gemma-S encoder can act as a general purpose code-to-metric regression model, by training over a large and diverse combination of regression data from GPU kernel programs, neural network architectures, and numerous different programming languages, as shown in Figure 1.

- Despite reading dense, complex ONNX representations for neural network graphs, RLMs are still able to remain competitive and even outperform state-of-the-art graph neural network (GNN)-based regression methods on standard neural architecture search (NAS) benchmarks. RLMs also naturally allow predicting multiple objectives such as latencies on different hardware.

- Comprehensive ablations demonstrate: (1) faster convergence curves when using model weights pretrained over standard language data and synthetic regression metrics, (2) decoder-based numeric outputs outperforming MSE-based regression heads, (3) improved regression with larger pretrained encoder sizes, and (4) important encoder settings such as tokenization and sequence length control.

Ultimately we hope this work paves the way for massively simplifying computational graph regression into a generic next-token prediction problem, aligning better with the modern large language model (LLM) paradigm.

## 2 RELATED WORK AND MOTIVATION

A fundamental issue of many previous techniques is the substantial effort required for feature engineering, when dealing with computational graphs. Even if a useful featurization can be found, typically the dependence on rigid aspects of the graph such as connectivity patterns and statistics may not be applicable to similar tasks, making them non-transferrable.

For example, in the compiler and programming languages communities, previous techniques (Nasr-Esfahany et al., 2025; Braberman et al., 2006; Akdere et al., 2012; Jayakumar et al., 2015; Johnston & Milthorpe, 2018) have proposed *count-based* techniques, by counting the occurrences of specific commands or aggregating program metrics and representing their statistics as a final fixed-length vector for tabular regression models such as multi-layer perceptrons (MLPs), random forests, and nearest neighbors. To align more with the graph-based nature of code, other works (Meng & Norris, 2017; Chennupati et al., 2021; Guan & Treude, 2024) first represent code as syntax trees over fixed corpuses of commands and then learn regression model coefficients over either specific components such as edges, or end-to-end via a GNN. Unfortunately, the moment a new command or kernel is introduced, this may invalidate all previous efforts and the entire process may need to be started from scratch.

Similar design patterns and issues exist for machine learning architectures, especially in field of NAS (White et al., 2023; Benmeziane et al., 2021; Elsken et al., 2019), where a key goal is to predict the performance of trained neural network-based computation graphs. Efforts have consisted of converting such graphs into tabular representations through the use of path encodings (White et al., 2021a), graph statistics (Kadlecová et al., 2024), zero-cost proxies (Abdelfattah et al., 2021) and activation information (Mellor et al., 2021). Other variants include creating graph kernels for the use in Gaussian Processes (Ru et al., 2021; Kandasamy et al., 2018) for Bayesian Optimization, and embeddings via the use of graph neural networks (Wen et al., 2020; Ning et al., 2020; Lukasik et al., 2021; White et al., 2021b; Akhauri & Abdelfattah, 2024b). To extend beyond the scope of purely predicting model accuracy but also latency and cost, additional techniques include hardware embeddings (Akhauri & Abdelfattah, 2023; 2024a; Lee et al., 2021), which require combining different features which have been processed by separate models.

Ideally, the use of minimally structured textual representations can ultimately resolve the issue of feature engineering, by sending strings directly to a single unified text-based regression model. However, such an idea has not yet gained wide popularity, presumably due to questions around their inductive bias, especially for high-precision code and graph regression problems. Nonetheless, there have been attempts (Qin et al., 2025; Zbinden et al., 2022) which attach regression heads to pretrained LLMs for NAS, and other attempts more broadly using LLMs for regression (Vacareanu et al., 2024; Lukasik et al., 2025) over tabular data and recommender systems. Our work crucially differs by establishing the general ability of language models to regress over many different code variants from pure text, which to the best of our knowledge has surprisingly not been investigated, yet is highly valuable for numerous computing fields.

## 3 METHOD

The RLM method can be taken standard from (Akhauri et al., 2025; Song et al., 2024), which fundamentally treats regression as a simple next-token prediction problem over $y$-values. The RLM is best structured as an encoder-decoder, which allows input representations of $x$ to be purely in text, taking advantage of the inherent flexibility of strings, and avoiding the need for one-hot representations of categories or normalization of numbers. One distinguishing aspect in this work is the use of a pretrained model (T5Gemma), which we show benefits code regression.

For the decoder side, it is best (as shown in Section 6.4) to use custom numeric tokenizations, such as the P10 tokenizer from (Charton, 2022) in which a $y$ is represented using special sign, mantissa, and exponent tokens, e.g. `<+><7><2><5><E-1>` represents $725 \times 10^{-1} = 72.5$. This tokenization is also normalization-free, avoiding numeric instabilities or the need to precompute minimum or maximum y-value bounds from data. At inference, constrained decoding is performed to ensure a valid number is always sampled, to either produce a pointwise prediction (via mean or median aggregation of samples) or density estimation with uncertainty quantification (Song & Bahri, 2025).

### 3.1 MULTI-TASK REGRESSION

Due to the universality of both the input and output representations, it is very straightforward to train $(x, y)$ data from multiple different regression tasks, which allows the use of a unified regression model. Furthermore, the RLM allows for a "pretrain then fine-tune" paradigm, where it can be pretrained on many real or even synthetic regression tasks, and then efficiently few-shot adapt to a new regression task via fine-tuning.

This is especially important as the string-based tokenization opens the doors for use on arbitrary string regression problems, but therefore may require more pretraining (either on regular language data or specific regression tasks) to understand specific structures such as low-level computation graphs better. Contrast this to a hand-crafted and heavily specialized graph regression model which can possess a better inductive bias for such problems, but whose use is restricted to only such formats. This can be more broadly seen as a consequence of the "no-free-lunch" theorem, where more universal methods require more data as they possess a larger space of hypotheses.

### 3.2 MULTI-OBJECTIVE MODELING

Due to the decoder's autoregressive nature, consecutively decoding more numbers also allows conditionally modeling multiple objectives $p(y' \mid y, x)$ which can naturally capture constraints inherent between different metrics. For example, if the latency ($y$) of a neural network is too low, the architecture may be too small and thus may not be possible to achieve a certain high level of image classification accuracy ($y'$). Previous works which rely on parallel regression heads at some embedding vector $\phi(x)$ are unable to capture correlations between metrics, as they make $y$ and $y'$ conditionally independent with respect to $\phi(x)$. We can generalize the conditional modeling to any number of metrics $k > 1$ via $p(y^{(k)} \mid y^{(k-1)}, \ldots, y^{(1)}, x)$, which we show in the experiments can be useful for predicting latencies across multiple hardware platforms.

## 4 DATA

### 4.1 HIGH-LEVEL PROGRAMMING DATASETS

We use several high level programming language datasets, to predict either the memory or execution latency from running the program on fixed hardware, as described in Table 1. Here, the texts align better with standard language model pretraining data.

APPS Leetcode: Hendrycks et al. (2021) contains 10K Python problems, with 232.4K ground-truth solutions and 131.7K test cases. We iterate over the `APPS` dataset, loading each solution and input-output pair, and run every solution in a minimal sandbox. Our primary metric is peak memory usage. We are able to successfully execute 99K solutions, with further details in Appendix D.4.1.

Triton Kernel Latency: KernelBook (Paliskara & Saroufim, 2025) pairs PyTorch programs with Triton kernels (example: Appendix D.5) produced by TorchInductor. We profile each Triton kernel's latency on a single NVIDIA A6000. Of the 18.2K problems, 12,652 kernels run successfully; most failures stem from our automated argument-matching harness rather than kernel correctness. Further details in Appendix D.4.2.

CodeNet: (Puri et al., 2021) introduces a large-scale dataset consisting of 14M code samples over 37 languages. We filter this dataset by "Accepted" solutions, resulting in 7.3M valid entries across several languages, and predict over the already provided memory column. Unfortunately, specific input program inputs are not provided, making it impossible to predict the memory *zero-shot* (i.e. new question, new submission). Nonetheless we can still evaluate the RLM on *limited information* scenarios since the train and test splits contain the same set of questions, allowing the RLM to still use few-shot submissions for a question during training, to infer on a submission for the same question at test time.

| Dataset | Samples | Languages | Problem Statement | Input Provided | Latency | Memory |
|---------|---------|-----------|-------------------|----------------|---------|--------|
| CodeNets | 7.39M | 37 | ✗ | ✗ | ✓ | ✓ |
| APPS | 98.9K | 1 | ✓ | ✓ | ✓ | ✓ |
| KernelBook | 12.6K | 1 | ✗ | ✓ | ✓ | ✗ |

Table 1: Coverage of high-level code datasets.

### 4.2 NAS DATASETS

In NAS, the primary objective is to predict the accuracy (e.g. on CIFAR-10) after training a neural network architecture with consistent hyperparameters. In this case our representation of choice is the Open Neural Network Exchange (ONNX) intermediate representation (IR) (ONNX Community, 2017), which contains full information about the auto-differentiation graph used, including all operations used and connectivity patterns. Unique to our work, the ONNX graph representation (example: Appendix D.6) is *universal* as it can represent any neural network or computation graph and is easily transferrable to any new possible neural network. It is also the default representation used in many ML compiler optimization efforts (Phothilimthana et al., 2023; Zheng et al., 2021; Kaufman et al., 2021), opening the doors to domains outside of purely NAS.

| Metric | Search space | | | | | | | | | |
|--------|------|--------|--------|------|--------|--------|--------|---------|-------|-----------|
| | NDS | NB-101 | NB-201 | FBNet | Ofa-MB | Ofa-PN | Ofa-RN | Twopath | Hiaml | Inception |
| Accuracy | ✓ | ✓ | ✓ | ✗ | ✓ | ✓ | ✓ | ✓ | ✓ | ✓ |
| Latency | ✗ | ✗ | ✓ | ✓ | ✗ | ✗ | ✗ | ✗ | ✗ | ✗ |
| Architectures | 44K | 423K | 15.6K | 5K | 7.5K | 8.2K | 10K | 6.9K | 4.6K | 580 |
| Median Tokens | 14K | 4.1K | 3.4K | 3.5K | 6.3K | 3.6K | 2.5K | 1.8K | 2.4K | 23K |

Table 2: Coverage of NAS metrics across search spaces.

Summarized in Table 2, we initialize and export all available architectures from NASBench-101 (Ying et al., 2019), NASBench-201 (Dong & Yang, 2020), FBNet (Wu et al., 2019), Once-for-all (Ofa)-MB/PN/RN (Cai et al., 2020), Twopath, Hiaml, Inception (Mills et al., 2023) and Network

Design Spaces (NDS) (Radosavovic et al., 2019) to a unified text-based ONNX IR. This amounts to a total of **520K** unique architectures represented in a unified format. We also collate their accuracy, FLOPs, parameter count and latencies. Further, we create our own NAS space (SNAS, see Appendix D.3) of 85.5K architectures, trained on CIFAR-10 for 32 steps, to serve as a pretraining space.

## 5 EXPERIMENTS

To demonstrate the simplicity of using a unified regressor, we jointly train our model on *all* of the training splits for the datasets mentioned above in Section 4. In Appendix A, we verify that despite absorbing very different forms of regression data (e.g. high-level code and ONNX graphs), the model's performance does not suffer. Appendix C contains exact hyperparameters used.

### 5.1 HIGH-LEVEL PROGRAMMING LANGUAGES

To begin, in Table 3, we find that the RLM produces non-trivial Spearman $\rho$ performances across multiple programming languages, with the strongest ($\rho > 0.9$) on APPS Leetcode peak-memory. On CodeNet, it performs the best on C++ but also remarkably decently on less common languages such as Lua and Haskell despite using such a small T5Gemma encoder, presumably pretrained minimally on more niche languages.

| Language | $\rho$ | Language | $\rho$ | Language | $\rho$ | Language | $\rho$ |
|---|---|---|---|---|---|---|---|
| C++ | 0.748 | Go | 0.670 | Python | 0.647 | Kotlin | 0.634 |
| C | 0.741 | D | 0.656 | OCaml | 0.643 | Swift | 0.630 |
| Lisp | 0.625 | Lua | 0.618 | Haskell | 0.611 | Rust | 0.611 |
| Perl | 0.592 | C# | 0.583 | Java | 0.560 | Scala | 0.537 |
| Fortran | 0.527 | TypeScript | 0.463 | Pascal | 0.461 | Ruby | 0.460 |
| Bash | 0.455 | F# | 0.439 | JavaScript | 0.395 | PHP | 0.347 |
| Triton Kernel Latency | | | | | | | 0.516 |
| APPS Leetcode Memory | | | | | | | 0.930 |

Table 3: Higher ($\uparrow$) is better. Evaluation on all high-level programming datasets, displaying Spearman $\rho$. We test 1024 programs per language. For CodeNet, we filter out languages which lack sufficient test examples, leading to 24 languages evaluated.

In Figure 2, we visualize $y$-values over different tasks and demonstrate the crucial design choice of our normalization-free $y$-representation, as the model is able to make predictions over a very wide range of scales, from $10^{-2}$ to $10^{6}$.

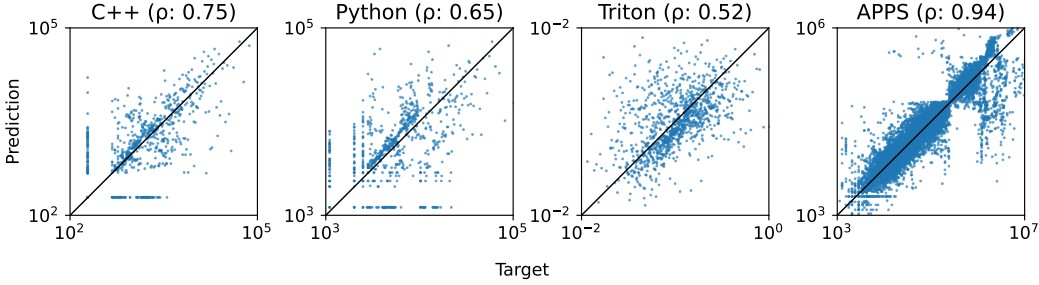

Figure 2: Diagonal fit ($\diagup$) is better. Scatterplot of RLM's pointwise $y$-prediction vs. ground truth value over varying tasks from CodeNet (C++ and Python), Triton Kernels, and APPS. For better visualization, axes are scaled by percentile (probits), and $y$-value ticks are shown at 10 and 90%.

Note that one substantial factor negatively influencing Spearman $\rho$ is the inherent flatness of $y$-values in some of the data in APPS, independent of the RLM. Using the RLM to rank solutions within a problem, we observed that the 5 problems with the worst performance also possess significantly

lower $y$-value spreads, with median coefficient of variation (CV) $\approx$ **0.0056** vs **0.037** (7x higher) than the 5 best problems. Furthermore, in Figure 3 (Left), we see that for more than half of problems, the RLM can achieve higher than $0.54$ Spearman $\rho$, and Figure 3 (Right) and additionally Figure 11 in Appendix B show the RLM can identify the best solution out of multiple submissions to a problem significantly better than random selection.

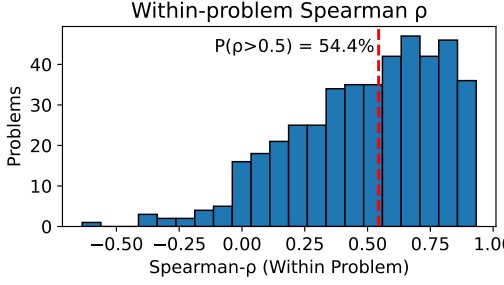
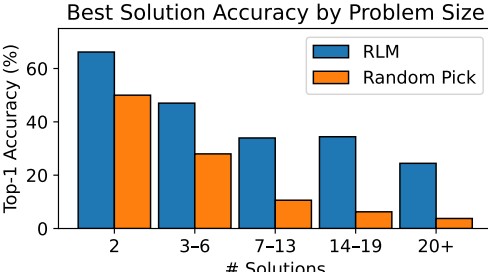

Figure 3: We identified problems with $>8$ candidate solution from our test set of 15000, and investigate whether the RLM is able to *rank* potential solutions. **(Left)** Distribution of problems and their in-problem Spearman $\rho$ rankings using the RLM. **(Right)** RLM vs random selection for choosing the top-1 lowest memory solution from a question, organized by solution count.

For qualitative inspection, in Figure 4 and Appendix D.7, we see that the RLM is able to distinguish memory consumption between two substantially different solutions for the same problem.

---

**Problem (Distance Value)**

Given two integer arrays `arr1` and `arr2`, and the integer `d`, return the distance value between the two arrays. The distance value is defined as the number of elements `arr1[i]` such that there is not any element `arr2[j]` where $|\text{arr1}[i] - \text{arr2}[j]| \le d$.

---

**Memory-efficient (O(1) extra space)**

```python
from typing import List

class Solution:
  def findTheDistanceValue(self, arr1:
    List[int], arr2: List[int], d:
    int) -> int:
    # O(1) memory.
    count = 0
    for a in arr1:
      far = True
      for b in arr2:
        # No allocations
        if abs(a - b) <= d:
          far = False
          # short-circuit
          break
      if far:
        count += 1
    return count
```

**Less memory-efficient (hash structures)**

```python
from typing import List
from collections import Counter

class Solution:
  def findTheDistanceValue(self, arr1:
    List[int], arr2: List[int], d:
    int) -> int:
    # overhead: builds dict
    arr1_counts = Counter(arr1)
    # overhead: build hash set
    arr2set    = set(arr2)

    total = 0
    for x in arr1_counts:
      target = range(x - d, x + d + 1)
      # overhead: new set
      if arr2set.intersection(target):
        continue
      total += arr1_counts[x]
    return total
```

Figure 4: Side-by-side solutions from the APPS dataset. Left minimizes memory (O(1) extra space, $O(nm)$ time). Right is often faster due to hash lookups but uses more memory via `Counter`, `set`, and per-iteration `intersection`. RLM predicted **5488** (left) and **10489.5** (right) bytes; ground truth: **5464** and **9672**.

## 5.2 NAS Results

In Figure 4, we further see that the RLM, consuming ONNX strings as input, remains competitive against even SoTA baselines such as FLAN (Akhauri & Abdelfattah, 2024b) and substantially outperforms other graph embedding techniques like Arch2Vec (Yan et al., 2020) which uses a graph autoencoder and CATE (Yan et al., 2021), which encodes architectures by feeding adjacency-matrix–derived token sequences into a Transformer to model global graph structure. We use these encodings with a MLP. Remarkably, the RLM does not require any additional information such as *zero-cost proxies* (Abdelfattah et al., 2021) which are crucial for FLAN to achieve strong results.

| Method | NASNet | Amoeba | PNAS | ENAS | DARTS | Average |
|---|---|---|---|---|---|---|
| MLP (Adjacency Enc.) | 0.002 | 0.032 | 0.082 | 0.021 | 0.124 | 0.052 |
| Arch2Vec (Graph Enc.) | 0.209 | 0.107 | 0.184 | 0.224 | 0.333 | 0.212 |
| CATE (Transformer Enc.) | 0.150 | 0.160 | 0.217 | 0.236 | 0.425 | 0.238 |
| GNN | 0.364 | 0.376 | **0.444** | 0.438 | 0.523 | 0.429 |
| FLAN$^T$ (Previous SoTA) | 0.344 | 0.470 | 0.430 | **0.484** | **0.567** | 0.459 |
| **RLM (Ours)** | **0.382** | **0.488** | 0.427 | 0.481 | 0.528 | **0.461** |

Table 4: Higher (↑) is better. Kendall $\tau$ rank correlation relative to prior SoTA (FLAN). We use 16 samples from the target search space for NASNet, Amoeba, PNAS and 100 samples for DARTS to match FLAN$^T$ settings. Note that MLP is trained from scratch due to different adjacency matrix sizes, while we use global representations of Arch2Vec and CATE.

In Figure 5, we further demonstrate the RLM's ability for multi-metric prediction, by assessing its decoder's ability to produce consecutive metrics. In addition to the accurate predicted Pareto-frontier, we also emphasize the slants of the densities, which demonstrate that the RLM decoder has inherently understood the positive correlation between architecture latency and accuracy, a benefit of its autoregressive design.

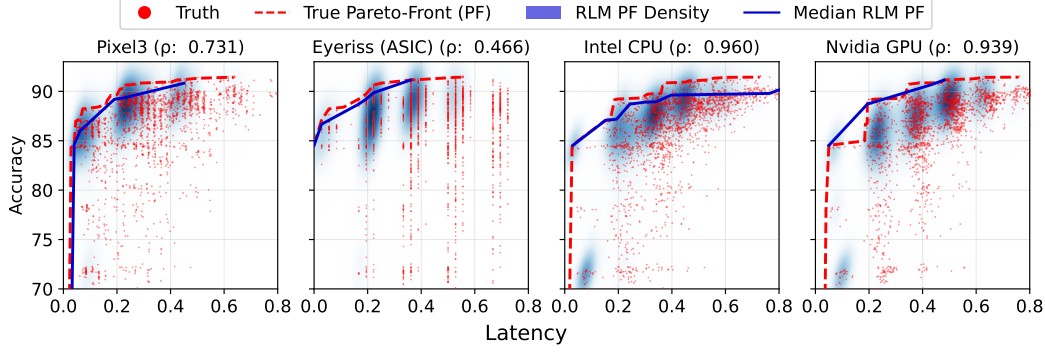

Figure 5: Single RLM trained on five consecutive objectives on NASBench-201, i.e. first validation accuracy and then hardware-specific latencies over four devices (Pixel3 (Mobile), Eyeriss (ASIC), Intel CPU and Nvidia GPU). Spearman $\rho$ refers to predicted latency. Density estimates (blue) are plotted for predicted Pareto-optimal points $x^*$.

## 6 Experiments: Ablations

### 6.1 Value of Pretraining

We ablate both notions of pretraining, i.e. (1) *language pretraining*: initializing from a (possibly frozen) encoder trained on language data, and (2) *regression pretraining:* initializing from scratch and training purely over (potentially synthetic) regression tasks. Note that these two are not in conflict, as one can still initialize from a language encoder while performing lots of further regression training.

In Figure 6, we see that a language pretrained model trains much better over the Triton Kernel task, leading to lower validation losses and subsequently better regression metrics. Freezing encoder does not impact our run, so we freeze it since its significantly cheaper.

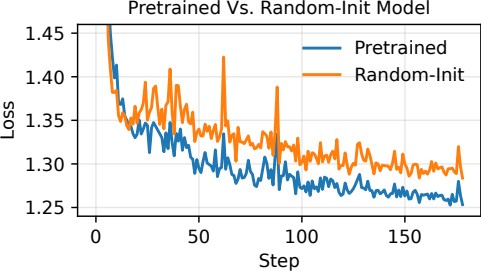

Figure 6: Lower (↓) is better. Validation loss curves when training from T5Gemma checkpoint (0.532 $\rho$) vs. random-init (0.504 $\rho$).

Figure 7: Lower (↓) is better. Validation loss curves when training from synthetic FLOPS pre-trained checkpoint (0.85 $\rho$) vs. random-init (0.83 $\rho$).

We further see the complementary value of regression pretraining, especially on cheap synthetic metrics. In Figure 8, we first show that RLMs can learn simple, synthetic metrics nearly perfectly, by pretraining on 381K NASBench-101 samples to predict floating point operations per second (FLOPS) for each architecture. We then re-initialize with this pretrained checkpoint and train over the real task of accuracy prediction from the exact same examples. This accelerates convergence and raises the final Spearman-$\rho$ as well, as shown in Figure 7.

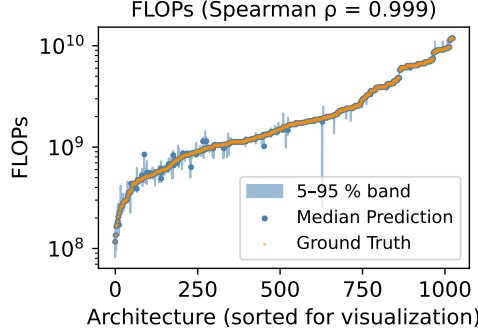

Figure 8: RLM predictions for FLOPS over 1024 test architectures.

## 6.2 COMPARING WITH REGRESSION HEADS

A common misconception is that performing regression with language models requires an explicit regression head (e.g., an MLP on pooled encoder states). To refute this, we use the same number of layers for fairness, and we compare an encoder–decoder (2 layers each) model trained with cross-entropy to an encoder-only (4 layers) model with an explicit regression head trained with mean squared error (MSE). We train these models on three NAS spaces, whose $y$-value ranges differ markedly (roughly 80–100 for NASBench-101, ∼50 for SNAS, and 0–1 for the OFA family).

Since MSE-based heads are sensitive to scale, we therefore evaluate two regression baselines: (i) *Regression Head* (no $y$-normalization) and (ii) *Normalized Regression Head* ($y$-values linearly scaled to $[0, 1]$ per dataset) used by Qin et al. (2025); Zbinden et al. (2022). In Table 5, normalization substantially improves the regression head (Spearman's $\rho = 0.717$ vs. $0.478$ without normalization), yet the decoder head remains best (Spearman's $\rho = 0.800$) and also has the practical advantage of being normalization-free across datasets.

## 6.3 SCALING REGRESSION LANGUAGE MODELS

(Akhauri et al., 2025) previously found that models trained from scratch, produce lower validation losses with increased parameter counts (up to 250M). To demonstrate scaling via pretrained models, we also train a 600M parameter model (HuggingFace: `t5gemma-b-b-prefixlm`) using the exact same settings as our default 300M model, and verify that it performs better (Table 6). However, we found that larger models in the T5Gemma family require extensive hyperparameter tuning and could not be run under limited compute - we leave further scaling analysis for future work.

| Head | Spearman-$\rho$ |
|------|-----------------|
| Regression Head | 0.478 |
| Normalized Regression Head | 0.717 |
| **Decoder Head (Ours)** | **0.800** |

Table 5: Higher ($\uparrow$) is better. Evaluations on 512 NASBench-101 test examples, using models pretrained on a subset of NASBench-101, SNAS, OfaRN, OfaPN, and OfaMB.

| T5Gemma | Params | Spearman-$\rho$ |
|---------|--------|-----------------|
| `s-s-prefixlm` | 300M | 0.744 |
| `b-b-prefixlm` | 600M | 0.782 |

Table 6: Higher ($\uparrow$) is better. Evaluations on 1024 CodeNet examples, using RLMs with different pretrained T5Gemma encoder sizes, trained on a smaller subset of CodeNet, APPS and KernelBook.

## 6.4 ENCODER-DECODER SETTINGS

Custom Encoder Tokenizations: We train an encoder–decoder from scratch and using SentencePiece (Kudo & Richardson, 2018) tokenization, compare using T5's default (32K tokens) (Raffel et al., 2020) to a custom, compact ONNX-aware tokenizer (8K tokens) learned from plain-text ONNX dumps. The learned tokenizer merges frequent operator strings (e.g., `MaxPool`) and reduces token counts, allowing longer graphs per sequence. This leads to a marked improvement in Table 7.

| | T5 (32K) | Learned (8K) |
|--|----------|--------------|
| Spearman-$\rho$ | 0.533 | **0.723** |

Table 7: Higher ($\uparrow$) is better. Spearman rank on 1024 test examples, when using default T5 vs. learned tokenizers and training on 381K NASBench-101 examples for one epoch.

| | 1K | 2K | 4K |
|--|----|----|----|
| RLM | 0.819 | 0.833 | 0.838 |

Table 8: Higher ($\uparrow$) is better. Sequence length ablation (Spearman-$\rho$) using learned encoder tokenizer on 381K NASBench-101 examples, for two epochs.

Longer Sequence Lengths: Using the learned tokenizer, increasing the encoder context allows the RLM to read more information about the graph, and thus improves rank correlation when using the same training procedure, with Spearman-$\rho$ rising from 0.819 (1K) to 0.838 (4K) in Table 8.

Decoder Tokenization and Initialization: The only change to the regular T5Gemma design is our use of P10 custom numeric tokenization with constrained decoding. To understand its effects, in Table 9, we see that the P10 tokenizer leads to better results against the regular T5Gemma tokenizer (i.e. 72.5 literally represented as `72.5`), as it induces better structuring on numbers and significantly simplifies decode token choices. Furthermore, using the pretrained T5Gemma decoder only

| | Decoder Initialization | |
|--------------|--------|------------|
| Tokenization | Random | Pretrained |
| P10 (Ours) | **0.747** | **0.744** |
| T5Gemma | 0.654 | 0.698 |

Figure 9: Higher is better ($\uparrow$). Evaluation on 1024 CodeNet samples after training. Note: P10 with pretrained decoder required resetting token embedding tables and final logit projection layer.

helps the T5Gemma tokenizer, presumably from relevant knowledge of numbers in common text format. However, P10 performance remains unchanged regardless of decoder pretraining, implying that only the T5Gemma pretrained encoder suffices for use.

## 7 CONCLUSION

Aligned with the standard generative pretraining paradigm (Radford et al., 2018), we have shown that RLMs are effective regression models for many types of programming languages and code representations, without requiring any post-processing or feature engineering of raw data. Applications include speeding up program search (Real et al., 2020; Romera-Paredes et al., 2024; Li et al., 2022), hardware-software co-design (Micheli & Gupta, 1997; Patterson & Hennessy, 2013), and compiler optimization (Wang & O'Boyle, 2018; Ashouri et al., 2018). A key open question is whether such code-based RLMs can be more broadly used to predict the numeric outcome of entire experiments from raw code, but we leave this to future work and hope this paper will be a valuable reference for multiple scientific communities in automated machine learning, programming languages, and computer architecture.

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

# A  UNIFIED MODEL ABLATIONS

## A.1  TRAINING ON CODE AND NAS

We verify below that training a unified model on both code and graphs does not harm its performance. In Table 9, a model trained with additional NAS graph data does *not* negatively impact ranking effectiveness (up to statistical significance) on any of the coding benchmarks, demonstrating that the RLM is able to absorb different domains.

| Pretrain Corpus | APPS (Py) | CN (C) | CN (C++) | CN (Py) | KernelBook | Avg. |
|---|---|---|---|---|---|---|
| Code | 0.942 | 0.684 | 0.741 | 0.651 | 0.486 | 0.701 |
| Code + Graphs | 0.925 | 0.740 | 0.733 | 0.634 | 0.499 | 0.706 |

Table 9: Higher (↑) is better. Spearman's $\rho$ values for an RLM trained only on code vs. an RLM trained additionally on NAS, when tested on coding benchmarks. We test on 1024 examples per language. CodeNet abbreviated as CN.

In Figure 10, we also see that throughout the training process, the validation Spearman $\rho$ does not change either, demonstrating consistent performance regardless of convergence.

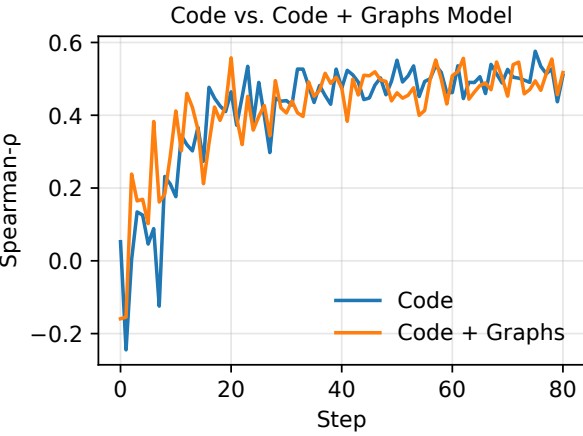

Figure 10: Higher is better (↑). Spearman $\rho$ on KernelBook examples, over different training checkpoints.

## A.2  PRETRAINING DIVERSITY AND IMPACT ON LANGUAGE

One important question is whether training on one language helps evaluation on other languages, as there may be some overlap in syntax or general programming styles. To study this using CodeNet, we fix the evaluation to always be over three languages (Go, Haskell, and Rust) while varying the pretraining mixture.

To remain fair, all models are trained over 482K examples, which always contains 45K fixed examples (15K from each language to be evaluated). The rest of the 437K examples are varied:

- One Language: C++ (437K samples)
- Two Languages: C++, Python (218.7K each)
- Four Languages: C++, Python, Java, Ruby (109.4K each)
- Six Languages: C++, Python, Java, Ruby, C#, C (72.9K each)

As shown in Table 10, increasing the number of pretraining languages does not clearly improve performance on unseen languages. For the three evaluation languages, the results stay roughly the

| Languages | In-Distribution | | | Purely Zero-Shot | | | | | | | | Average |
|---|---|---|---|---|---|---|---|---|---|---|---|---|
| | Go | Haskell | Rust | D | Fortran | JavaScript | Kotlin | OCaml | PHP | Perl | Scala | |
| One | 0.61 | 0.56 | 0.57 | 0.53 | 0.38 | 0.29 | 0.49 | 0.58 | 0.33 | 0.23 | 0.40 | 0.45 |
| Two | 0.62 | 0.57 | 0.57 | 0.56 | 0.30 | 0.34 | 0.54 | 0.60 | 0.24 | 0.17 | 0.40 | 0.45 |
| Four | 0.61 | 0.51 | 0.56 | 0.55 | 0.28 | 0.25 | 0.39 | 0.52 | 0.12 | 0.05 | 0.25 | 0.37 |
| Six | 0.61 | 0.54 | 0.55 | 0.52 | 0.34 | 0.31 | 0.44 | 0.57 | 0.27 | 0.17 | 0.37 | 0.43 |

Table 10: Higher ($\uparrow$) is better. Spearman $\rho$ results across languages.

same across all settings. For purely zero-shot languages that the model never saw during training (e.g. D, Fortran, ...), the increased pretraining diversity even sometimes leads to worse results.

We hypothesize this occurs because of the structure of the CodeNet dataset, which contains 13,916,868 submissions divided across 4053 problems. In practice, seeing more diverse problems in a single language may be more helpful than seeing the same problems repeated across multiple languages. In other words, the model benefits more from variety in problem content than from variety in programming syntax. This effect may be reinforced by the strong T5Gemma encoder, which already encodes different programming languages well, making additional cross-language diversity less important.

## A.3  FINE-TUNING

In Table 11, we further show that even fine-tuning on data from a specific language, does *not* necessarily help its performance when the task was already richly observed from the pretraining corpus. We hypothesize this is a form of "catastrophic forgetting", where over-focusing on a specific language can actually negatively affect general reasoning and regression abilities, driving the overall result down. Furthermore, T5Gemma encoder is already well-calibrated for code, and thus the benefit of fine-tuning with just 1024 samples may be relatively limited.

| | C++ | C | Go | Python | Rust | Haskell | C# | Java | Ruby | Triton |
|---|---|---|---|---|---|---|---|---|---|---|
| No FT | 0.730 | 0.714 | 0.655 | 0.637 | 0.607 | 0.577 | 0.538 | 0.518 | 0.450 | 0.501 |
| FT | 0.595 | 0.569 | 0.639 | 0.448 | 0.566 | 0.546 | 0.472 | 0.452 | 0.335 | 0.492 |

Table 11: Higher ($\uparrow$) is better. Spearman $\rho$ performance of models with and without fine-tuning (FT) across different programming languages. The model is pretrained on a sufficiently large corpus of code, and does not benefit from 1024 new few-shot examples specific to the language being evaluated. We test 1024 programs per language.

For NAS however, fine-tuning *does* benefit performance on out-of-domain tasks. In Table 12, we took our pretrained model on both code and NAS, and fine-tune it an an additional 1K samples from the target NAS search space. While Amoeba and ENAS were in the pretraining set, they were only 0.08% of the pretraining corpus, while the total NAS data also only occupied 1.1%. Thus for such low-resource tasks, there is significant benefit to fine-tuning the RLM, leading to the massive gains (+0.35 Spearman $\rho$ on Amoeba and ENAS).

| | NASBench201 | NASBench101 | ENAS | Amoeba |
|---|---|---|---|---|
| No FT | 0.681 | 0.646 | 0.165 | 0.045 |
| FT | 0.738 | 0.734 | 0.516 | 0.501 |

Table 12: Higher ($\uparrow$) is better. Spearman's $\rho$ performance of models with and without fine-tuning (FT) on NAS. We test 1024 architectures for search space.

# B ADDITIONAL EXPERIMENTS

## B.1 LIMITED INFORMATION SCENARIO

As mentioned in Section 4, despite the CodeNet dataset not displaying inputs to the code submissions, it is still possible to predict memory consumption via shared questions from both training and test time. We demonstrate this is also the case for APPS in Table 13, where omitting the problem statement (containing input information) does not significantly harm predictions (only a drop of $0.08\,\rho$) for code latency.

| RLM Input | Spearman $\rho$ |
|---|---|
| Problem + Code (Default) | 0.93 |
| Code Only | 0.85 |

Table 13: Higher ($\uparrow$) is better. Spearman $\rho$ for when the model is trained over problem and code (default setting), vs. observing the code submission only. We test 1024 programs per language.

## B.2 RANKING

Continuing from Figure 3, we also provide further evidence that the RLM is capable of selecting the lowest latency (i.e. fastest) code submissions for a given question on APPS. In many cases, top-1 identification can be impossible as there are numerous submissions with very similar or identical implementations. For example, one maximum-subarray question in APPS has 4 out of 20 submissions using exactly the same "Kadane's algorithm" (Bentley, 1984). Instead, we vary the top-x% in Figure 11, to show that the RLM can at least identify the top percentile of submissions in general.

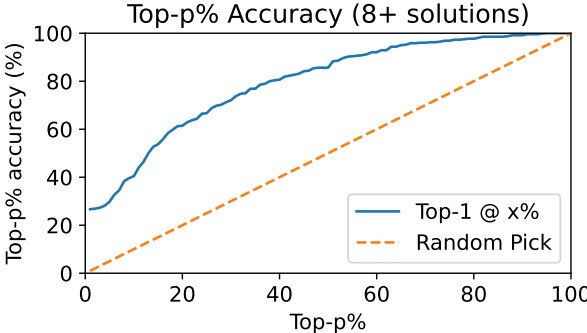

Figure 11: Higher ($\uparrow$) is better. Fraction of problems (with $>8$ solutions) where the model's predicted best solution lies within the true top-$p\%$ of solutions; dashed line shows the random pick baseline.

## C  EXPERIMENTAL SETTINGS

We use the codebase from `https://github.com/google-deepmind/regress-lm` to train the RLM. We use the following default hyperparameters:

- **Optimization and schedule.** We use Adafactor. Pretraining uses a learning rate of $1\times10^{-3}$; fine-tuning uses $5\times10^{-5}$. Gradients are clipped at a global norm of $2.0$. The scheduler is a linear warmup for the first $10\%$ of steps followed by cosine decay.

- **Decoder sizes:** We match the corresponding T5Gemma model where mentioned. Otherwise, we use two decoder layers, with hidden-sizes 2048 for both attention (with 8 heads) and feedforward.

- **Inference:** We take the median of 64 samples from the decoder for our pointwise estimate. The sample size can be increased to produce even more accurate pointwise predictions, but we found this default was sufficient.

- **Input length:** Our encoder uses a maximum of 2048 token lengths, and crops any tokenization sequences beyond this limit. Truncation only occurred for ONNX graphs from NAS data, but this does not significantly harm performance (as seen in Table 8) as cell structures repeat throughout the architecture.

# D   DATA: EXTENDED

## D.1   $y$-VALUE DISTRIBUTIONS

In Figure 12, we plot the histogram of all $y$-values encountered in the datasets. This is to demonstrate the wildly different value ranges both across and within datasets, ranging from $10^{-1}$ to $10^5$ orders of magnitude. We emphasize that these ranges would make training using an MSE-based loss incredibly difficult, due to the sheer amount of variability of per-example loss magnitudes, and tedious normalizations to be performed per dataset.

This further highlights the necessity and benefit of using (1) cross-entropy as the loss for each example is well-behaved and (2) decoder head which does not require any $y$-normalizations.

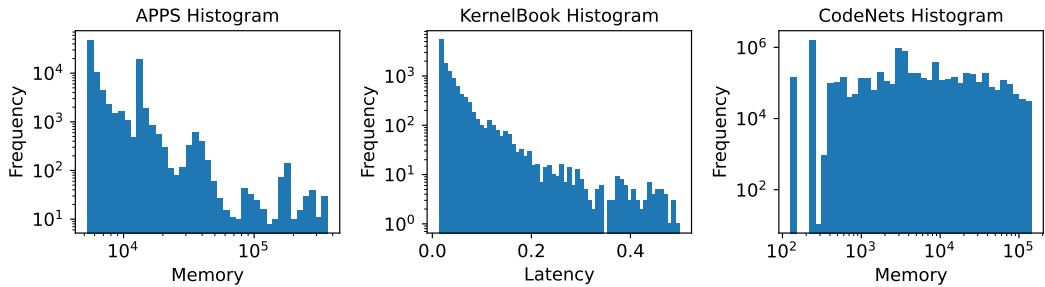

Figure 12: Histogram of the target values for APPS, KernelBook and CodeNet

## D.2   EXTRA LEETCODE DATA FOR APPS

As a small aside, in APPS, we also appended additional 600 examples from `EffiBench` (Huang et al., 2025), another set of Leetcode problems and submissions. For each problem, `generate_test_case()` provides the inputs and expected output, and we measure the wall-clock time of repeatedly running the solution on these cases, averaging over many iterations and trials.

## D.3   GENERATING THE SNAS DATASET

We construct the SNAS dataset by repeatedly sampling, briefly training, and recording lightweight CNN architectures on CIFAR-10 under a fixed-budget protocol:

- **Sampling.** For each example, we draw a macro configuration (e.g., stem width, stacks, cells per stack, width multiplier) and a micro cell DAG with operations from a small registry; residual connections may be enabled. The resulting network is serialized to a compact, reconstructable `arch_str`.

- **Training & evaluation.** Each sampled network is trained for a small, fixed budget (steps or wall time) using SGD with momentum and a cosine learning-rate schedule under mixed precision (FP16/BF16). Augmentation and normalization follow standard CIFAR-10 practice and are executed on-GPU (8 Nvidia A6000 and 5 3090 GPUs). We report top-1 accuracy on a held-out evaluation subset of the test split.

- **Logging.** We stream one JSONL record per architecture with `uid`, `val_accuracy` (primary label), `params`, `train_time_sec`, `steps_ran`, `precision`, `batch_size`, and `arch_str`.

## D.4   HARDWARE PROFILING

Below, we discuss specific details on how we collected $y$-values for varying code datasets.

### D.4.1 APPS

We use the following system configuration to profile problems from the APPS Hendrycks et al. (2021) dataset.

- **CPU:** AMD EPYC 7702 ("Rome"), 1× socket, 64 cores / 128 threads (SMT enabled); boost enabled; frequency range ∼1.50–2.18 GHz.
- **Topology & Caches:** L1d: 2 MiB total (64 instances); L1i: 2 MiB total (64 instances); L2: 32 MiB total (64 instances); L3: 256 MiB total (16 slices).
- **NUMA:** Single node (node0 CPUs 0–127).
- **Memory:** 503 GiB RAM (no swap configured).
- **OS/Kernel:** Ubuntu 22.04, Linux `6.8.0-45-generic` (x86_64).

We profile Python solutions from the APPS train split with a small wrapper and consistent run protocol; the primary metric is `dyn_peak_alloc_bytes`.

- For each problem, load `solutions.json` and `input_output.json`.
- **Execution modes.** If `fn_name` exists, run in *callable* mode by passing JSON args; otherwise run as *stdin* program. Each run executes in a fresh Python process with `-I -S -B`.
- **Wrapper basics.** Pre-import common stdlib modules, raise recursion limit, keep site-packages importable under `-I/-S`, set `PYTHONHASHSEED=0`. Outputs are discarded during timing.
- **Warmup & repeats.** Per (solution,input): discard `warmup` runs (default 3), then measure `repeats` (default 11). Per-run timeout: 10s.
- **Timing.** *Wall time*: `perf_counter_ns`. *CPU time* (POSIX): RUSAGE_CHILDREN deltas.
- **Dynamic memory (primary).** Via `tracemalloc`, one untimed instrumented run per solution collects `dyn_peak_alloc_bytes`, `dyn_alloc_bytes_pos`, and `dyn_alloc_count_pos` (attributed to the user file). One `ru_maxrss` collects `dyn_rss_peak_bytes`. Lightweight trace/profile counters (line events, call count, max depth) are also recorded.
- **Output.** We write one CSV row per (solution,input set) with summary stats (min/median/mean/p90/max/stddev/variance) for wall and CPU time, run counts, Python version, host, UTC timestamp, and the dynamic metrics above.

We report `dyn_peak_alloc_bytes` (from tracemalloc) as our primary memory metric because it isolates Python-heap usage; peak RSS (`dyn_peak_alloc_bytes`) is provided as a secondary, noisier indicator capturing native allocations (e.g., NumPy) and allocator effects. This emphasizes Python-level memory complexity while still flagging cases dominated by non-Python memory. Our target for the RLM is `dyn_peak_alloc_bytes`.

### D.4.2 KERNELBOOK

We use the following system configuration for KernelBook Paliskara & Saroufim (2025) A6000 profiling.

- **CPU:** Intel Xeon Gold 6448Y, 2× sockets, 64 cores / 64 threads (SMT disabled); boost enabled; frequency range ∼0.80–2.10 GHz.
- **Topology & Caches:** L1d: 3 MiB total (64 instances); L1i: 2 MiB total (64 instances); L2: 128 MiB total (64 instances); L3: 120 MiB total.
- **NUMA:** Two nodes (node0 CPUs 0–31; node1 CPUs 32–63).
- **Memory:** 1008 GiB RAM; 4 GiB swap.
- **GPU/Driver:** 1× NVIDIA RTX A6000 (48 GiB), driver 530.30.02; CUDA 12.1.

We profile each Triton kernel from KernelBook on a single NVIDIA A6000. After a short JIT warmup, we time an adaptive loop seeded at 20 iterations and extended to a $\geq 1\,$s window; this window is repeated for 5 trials. We report median latency (ms) and also record the across-trial standard deviation.

For inputs, we use the dataset-provided constructors and activations, automatically trying a small set of argument orderings (parameters first, activations first, and interleavings) and using the first that passes shape checks. Per kernel, we write `[index, sha, latency_ms, stddev_ms]` to CSV and continue past failures (e.g., OOM or shape mismatch) without aborting the run.

1134
1135
1136
1137
1138

## D.5 TRITON CODE SAMPLE (KERNELBOOK)

1139

**Triton Kernel Example #4949; Latency (0.0152 ms)**

1140
1141
1142
1143
1144
1145
1146
1147
1148
1149
1150
1151
1152
1153
1154
1155
1156
1157
1158
1159
1160
1161
1162
1163
1164
1165
1166
1167
1168
1169
1170
1171
1172
1173
1174
1175
1176
1177
1178
1179
1180
1181
1182
1183
1184
1185
1186
1187

```python
import torch
import triton
import triton.language as tl
from torch._inductor.runtime.triton_heuristics import grid
from torch._C import _cuda_getCurrentRawStream as get_raw_stream
from torch._inductor.runtime import triton_helpers
from torch import nn
assert_size_stride = torch._C._dynamo.guards.assert_size_stride
empty_strided_cuda = torch._C._dynamo.guards._empty_strided_cuda

@triton.jit
def triton_per_fused_add_div_mul_rsub_sum_0(in_out_ptr0, in_ptr0, in_ptr1,
    xnumel, rnumel):
    XBLOCK: tl.constexpr = 1
    RBLOCK: tl.constexpr = 256
    xoffset = tl.program_id(0) * XBLOCK
    tl.full([1], xoffset, tl.int32)
    tl.full([RBLOCK], True, tl.int1)
    rindex = tl.arange(0, RBLOCK)[:]
    tl.full([RBLOCK], True, tl.int1)
    r0 = rindex
    tmp0 = tl.load(in_ptr0 + r0, None)
    tmp1 = tl.load(in_ptr1 + r0, None)
    tmp2 = tmp0 * tmp1
    tmp3 = tl.broadcast_to(tmp2, [RBLOCK])
    tmp5 = triton_helpers.promote_to_tensor(tl.sum(tmp3, 0))
    tmp6 = tl.broadcast_to(tmp0, [RBLOCK])
    tmp8 = triton_helpers.promote_to_tensor(tl.sum(tmp6, 0))
    tmp9 = tl.broadcast_to(tmp1, [RBLOCK])
    tmp11 = triton_helpers.promote_to_tensor(tl.sum(tmp9, 0))
    tmp12 = 2.0
    tmp13 = tmp5 * tmp12
    tmp14 = 1.0
    tmp15 = tmp13 + tmp14
    tmp16 = tmp8 + tmp11
    tmp17 = tmp16 + tmp14
    tmp18 = tmp15 / tmp17
    tmp19 = tmp14 - tmp18
    tl.debug_barrier()
    tl.store(in_out_ptr0 + tl.full([1], 0, tl.int32), tmp19, None)

def call(args):
    arg0_1, arg1_1 = args
    args.clear()
    assert_size_stride(arg0_1, (4, 4, 4, 4), (64, 16, 4, 1))
    assert_size_stride(arg1_1, (4, 4, 4, 4), (64, 16, 4, 1))
    with torch.cuda._DeviceGuard(0):
        torch.cuda.set_device(0)
        buf0 = empty_strided_cuda((), (), torch.float32)
        buf3 = buf0
        del buf0
        get_raw_stream(0)
        triton_per_fused_add_div_mul_rsub_sum_0[grid(1)](buf3, arg0_1,
            arg1_1, 1, 256, num_warps=2, num_stages=1)
        del arg0_1
        del arg1_1
    return buf3,

class DiceLossNew(nn.Module):

    def __init__(self, weight=None, size_average=True):
        super(DiceLossNew, self).__init__()

    def forward(self, input_0, input_1):
        arg0_1 = input_0
        arg1_1 = input_1
        output = call([arg0_1, arg1_1])
        return output[0]
```

## D.6 ONNX GRAPH CODE SAMPLE

---

**ONNX Graph (SNAS Architecture #10. Accuracy: 60.93%)**

```
graph main_graph (  %input[FLOAT, 1x3x32x32] %features.0.conv.weight[FLOAT, 16x3x3x3]  %↩
    ↪ features.0.bn.weight[FLOAT, 16]  %features.0.bn.bias[FLOAT, 16]  %features.0.bn.↩
    ↪ running_mean[FLOAT, 16]  %features.0.bn.running_var[FLOAT, 16]  %features.1.ops.1↩
    ↪ .op.1.weight[FLOAT, 6x1x7x7]  %features.1.ops.1.op.2.weight[FLOAT, 6x6x1x1]  %↩
    ↪ features.1.ops.1.op.3.weight[FLOAT, 6]  %features.1.ops.1.op.3.bias[FLOAT, 6]  %↩
    ↪ features.1.ops.1.op.3.running_mean[FLOAT, 6]  %features.1.ops.1.op.3.running_var[↩
    ↪ FLOAT, 6]  %features.1.ops.1.op.5.weight[FLOAT, 6x1x7x7]  %features.1.ops.1.op.6.↩
    ↪ weight[FLOAT, 6x6x1x1]  %features.1.ops.1.op.7.weight[FLOAT, 6]  %features.1.ops.↩
    ↪ 1.op.7.bias[FLOAT, 6]  %features.1.ops.1.op.7.running_mean[FLOAT, 6]  %features.1↩
    ↪ .ops.1.op.7.running_var[FLOAT, 6]  %features.1.ops.2.conv.weight[FLOAT, 6x6x1x1] ↩
    ↪  %features.1.ops.2.bn.weight[FLOAT, 6]  %features.1.ops.2.bn.bias[FLOAT, 6]  %↩
    ↪ features.1.ops.2.bn.running_mean[FLOAT, 6]  %features.1.ops.2.bn.running_var[↩
    ↪ FLOAT, 6]  %features.1.ops.3.conv.conv.weight[FLOAT, 6x6x3x3]  %features.1.ops.3.↩
    ↪ conv.bn.weight[FLOAT, 6]  %features.1.ops.3.conv.bn.bias[FLOAT, 6]  %features.1.↩
    ↪ ops.3.conv.bn.running_mean[FLOAT, 6]  %features.1.ops.3.conv.bn.running_var[FLOAT↩
    ↪ , 6]  %features.1.ops.4.op.1.weight[FLOAT, 5x1x3x3]  %features.1.ops.4.op.2.↩
    ↪ weight[FLOAT, 5x5x1x1]  %features.1.ops.4.op.3.weight[FLOAT, 5]  %features.1.ops.↩
    ↪ 4.op.3.bias[FLOAT, 5]  %features.1.ops.4.op.3.running_mean[FLOAT, 5]  %features.1↩
    ↪ .ops.4.op.3.running_var[FLOAT, 5]  %features.1.ops.4.op.5.weight[FLOAT, 5x1x3x3]

%%%%%%%%%%%%%%%%%%%%%%%%%%%% Code Ommited For Brevity %%%%%%%%%%%%%%%%%%%%%%%%%%%%%
%%%%%%%%%%%%%%%%%%%%%%%%%%%% Code Ommited For Brevity %%%%%%%%%%%%%%%%%%%%%%%%%%%%%

%/features/features.8/ops.1/act/Relu_output_0 = Relu(%/features/features.8/ops.1/op/Max↩
    ↪ Pool_output_0)  %/features/features.8/ops.2/op/op.0/Relu_output_0 = Relu(%/↩
    ↪ features/features.8/ops.1/act/Relu_output_0)  %/features/features.8/ops.2/op/op.1↩
    ↪ /Conv_output_0 = Conv[dilations = [1, 1], group = 32, kernel_shape = [3, 3], pads↩
    ↪  = [1, 1, 1, 1], strides = [1, 1]](%/features/features.8/ops.2/op/op.0/↩
    ↪ Relu_output_0, %features.8.ops.2.op.1.weight)  %/features/features.8/ops.2/op/op.↩
    ↪ 2/Conv_output_0 = Conv[dilations = [1, 1], group = 1, kernel_shape = [1, 1], pads↩
    ↪  = [0, 0, 0, 0], strides = [1, 1]](%/features/features.8/ops.2/op/op.1/↩
    ↪ Conv_output_0, %features.8.ops.2.op.2.weight)  %/features/features.8/ops.2/op/op.↩
    ↪ 3/BatchNormalization_output_0 = BatchNormalization[epsilon = 9.99999974737875e-06↩
    ↪ , momentum = 0.899999976158142](%/features/features.8/ops.2/op/op.2/Conv_output_0↩
    ↪ , %features.8.ops.2.op.3.weight, %features.8.ops.2.op.3.bias, %features.8.ops.2.↩
    ↪ op.3.running_mean, %features.8.ops.2.op.3.running_var)  %/features/features.8/ops↩
    ↪ .2/op/op.4/Relu_output_0 = Relu(%/features/features.8/ops.2/op/op.3/↩
    ↪ BatchNormalization_output_0)  %/features/features.8/ops.2/op/op.5/Conv_output_0 =↩
    ↪  Conv[dilations = [1, 1], group = 32, kernel_shape = [3, 3], pads = [1, 1, 1, 1],↩
    ↪  strides = [1, 1]](%/features/features.8/ops.2/op/op.4/Relu_output_0, %features.8↩
    ↪ .ops.2.op.5.weight)  %/features/features.8/ops.2/op/op.6/Conv_output_0 = Conv[↩
    ↪ dilations = [1, 1], group = 1, kernel_shape = [1, 1], pads = [0, 0, 0, 0], ↩
    ↪ strides = [1, 1]](%/features/features.8/ops.2/op/op.5/Conv_output_0, %features.8.↩
    ↪ ops.2.op.6.weight)  %/features/features.8/ops.2/op/op.7/↩
    ↪ BatchNormalization_output_0 = BatchNormalization[epsilon = 9.99999974737875e-06, ↩
    ↪ momentum = 0.899999976158142](%/features/features.8/ops.2/op/op.6/Conv_output_0, ↩
    ↪ %features.8.ops.2.op.7.weight, %features.8.ops.2.op.7.bias, %features.8.ops.2.op.↩
    ↪ 7.running_mean, %features.8.ops.2.op.7.running_var)  %/features/features.8/↩
    ↪ input_proj.3/conv/Conv_output_0 = Conv[dilations = [1, 1], group = 1, ↩
    ↪ kernel_shape = [1, 1], pads = [0, 0, 0, 0], strides = [1, 1]](%/features/features↩
    ↪ .7/Concat_output_0, %features.8.input_proj.3.conv.weight)  %/features/features.8/↩
    ↪ input_proj.3/bn/BatchNormalization_output_0 = BatchNormalization[epsilon = 9.9999↩
    ↪ 9974737875e-06, momentum = 0.899999976158142](%/features/features.8/input_proj.3/↩
    ↪ conv/Conv_output_0, %features.8.input_proj.3.bn.weight, %features.8.input_proj.3.↩
    ↪ bn.bias, %features.8.input_proj.3.bn.running_mean, %features.8.input_proj.3.bn.↩
    ↪ running_var)  %/features/features.8/ops.3/op/AveragePool_output_0 = AveragePool[↩
    ↪ ceil_mode = 0, count_include_pad = 0, kernel_shape = [3, 3], pads = [1, 1, 1, 1],↩
    ↪  strides = [1, 1]](%/features/features.8/input_proj.3/bn/↩
    ↪ BatchNormalization_output_0)  %/features/features.8/ops.3/act/Relu_output_0 = ↩
    ↪ Relu(%/features/features.8/ops.3/op/AveragePool_output_0)  %/features/features.8/↩
    ↪ Add_output_0 = Add(%/features/features.8/ops.2/op/op.7/BatchNormalization_output_↩
    ↪ 0, %/features/features.8/ops.3/act/Relu_output_0)  %/features/features.8/ops.4/op↩
    ↪ /AveragePool_output_0 = AveragePool[ceil_mode = 0, count_include_pad = 0, ↩
    ↪ kernel_shape = [3, 3], pads = [1, 1, 1, 1], strides = [1, 1]](%/features/features↩
    ↪ .8/Add_output_0)  %/features/features.8/ops.4/act/Relu_output_0 = Relu(%/features↩
    ↪ /features.8/ops.4/op/AveragePool_output_0)  %/features/features.8/Concat_output_0↩
    ↪  = Concat[axis = 1](%/features/features.8/ops.3/act/Relu_output_0, %/features/↩
    ↪ features.8/ops.4/act/Relu_output_0)  %/GlobalAveragePool_output_0 = ↩
    ↪ GlobalAveragePool(%/features/features.8/Concat_output_0)  %/Flatten_output_0 = ↩
    ↪ Flatten[axis = 1](%/GlobalAveragePool_output_0)  %logits = Gemm[alpha = 1, beta =↩
    ↪  1, transB = 1](%/Flatten_output_0, %classifier.weight, %classifier.bias)  return↩
    ↪  %logits}
```

## D.7 EXAMPLE CODE SUBMISSIONS

---

**Problem (Maximum Subarray Sum with One Deletion)**

Given an integer array `arr`, return the maximum sum of a non-empty subarray after option-ally deleting at most one element from that subarray (the result must still be non-empty).

---

**Less memory-efficient**

```python
from typing import List

class Solution:
  def maximumSum(self, arr: List[int])
      -> int:
    # extra memory overhead
    max_res   = [0] * len(arr)
    max_start = [0] * len(arr)
    max_end   = [0] * len(arr)

    for i, n in enumerate(arr):
      max_end[i] = n if i == 0 else
     max(n, max_end[i-1] + n)
    # debug overhead
    print(max_end)
    # materialize reverse pass array
    for i, n in list(enumerate(arr))
     [::-1]:
      max_start[i] = n if i == len(arr
     ) - 1 else max(n, max_start[i+1] +
     n)
    # debug overhead
    print(max_start)

    for i, n in enumerate(arr):
      left  = n if i == 0 else max_end
     [i-1]
      right = n if i == len(arr) - 1
     else max_start[i+1]
      max_res[i] = max(left, right,
     left + right)
    # debug overhead
    print(max_res)

    return max(max_res)
```

**Memory-efficient (O(1) extra space)**

```python
from typing import List

class Solution:
  def maximumSum(self, arr: List[int])
      -> int:
    # keep: best sum with no deletion
    # drop: best sum with one deletion
    keep = arr[0]
    drop = float('-inf')
    ans  = arr[0]

    for x in arr[1:]:
      # delete current x OR already
     deleted
      drop = max(drop + x, keep)
      # Kadane
      keep = max(keep + x, x)
      ans  = max(ans, keep, drop)

    return ans
```

Figure 13: LeetCode "Maximum Subarray Sum with One Deletion". **(Left)**: one-pass DP that keeps only two running states (`keep`, `drop`)—$\mathcal{O}(1)$ extra space and $\mathcal{O}(n)$ time. **(Right)**: builds three length-$n$ arrays (`max_end`, `max_start`, `max_res`)—$\mathcal{O}(n)$ extra space and $\mathcal{O}(n)$ time. *Ground truth memory:* $\mathcal{O}(1)$ version **5608** bytes; less memory-efficient version **7136** bytes. *RLM predictions:* **5549** and **6228** bytes, respectively. The gap comes from (i) storing three auxiliary arrays of size $n$, (ii) materializing `list(enumerate(arr))` for the reverse pass, and (iii) debug `print(...)` calls that create large temporary strings when printing full arrays.

---

### Problem (Maximum Sum Circular Subarray)

Given an integer array `A` that represents a circular array, return the maximum possible sum of a non-empty subarray of the circular array. Wrap-around is allowed, but each element of the fixed buffer `A` may be used at most once in the subarray.

---

**Memory-efficient (O(1) extra space)**

```python
from typing import List

class Solution:
  def maxSubarraySumCircular(self, A:
    List[int]) -> int:
    # scalar accumulators O(1) space
    maxsum = minsum = A[0]
    # rolling state; no array alloc
    curmax = curmin = total = 0

    for num in A:
      # Kadane step for max
      curmax = max(num, curmax + num)
      maxsum = max(maxsum, curmax)

      # Min-Kadane
      curmin = min(num, curmin + num)
      minsum = min(minsum, curmin)

      # single pass accumulation
      total += num
    # computed from scalars; (O(1))
    return max(maxsum, total - minsum)
      if maxsum > 0 else maxsum
```

**Less memory-efficient (O(n) extra space)**

```python
from typing import List

class Solution:
  def maxSubarraySumCircular(self, A:
    List[int]) -> int:

    def maxSubarray(A: List[int]) ->
    int:
      # length-n DP array (O(n))
      dp = [0] * len(A)
      dp[0] = A[0]
      for i in range(1, len(A)):
        dp[i] = max(A[i], dp[i - 1] +
    A[i])
      # entire DP history to take max
      return max(dp)

    temp = maxSubarray(A)

    res = float('-inf')
    # allocates rightMax (length-n)
    rightMax = [max(A[0], 0)] + [0] *
    (len(A) - 1)
    currMax  = max(A[0], 0)
    # prefix-tracking with rightWin
    rightWin = [A[0]] + [0] * (len(A)
    - 1)

    for idx, x in enumerate(A[1:]):
      currMax       = max(x + rightWin
    [idx], currMax)
      rightMax[idx+1] = currMax
      rightWin[idx+1] = x + rightWin[
    idx]
    # extra full pass
    A.reverse()
    # suffix-sum with leftWin
    leftWin = [A[0]] + [0] * (len(A) -
    1)
    for idx, x in enumerate(A[1:]):
      leftWin[idx + 1] = x + leftWin[
    idx]
      currMax = rightMax[len(A) - idx
    - 2]
      res = max(res, currMax + leftWin
    [idx])

    return max(res, temp)
```

Figure 14: Side-by-side solutions for the circular maximum subarray problem. Both run in $O(n)$ time. **(Left) (O(1) space)** tracks only scalar accumulators (Kadane for max and min + total), which avoids auxiliary arrays. **(Right) (O(n) space)** allocates several length-$n$ arrays (`dp`, `rightMax`, `rightWin`, `leftWin`) and also reverses `A` in place, increasing memory usage. RLM memory footprint estimated: **5634** (left, more memory-efficient) vs. **6430** (right, less memory-efficient). (Ground truth: **5508** and **6528**, respectively.)

**Problem (Array of Doubled Pairs)**

Given an integer array $A$ of even length, return `True` iff it is possible to reorder it so that $A[2i+1] = 2 \cdot A[2i]$ for every $0 \le i < |A|/2$.

**Constraints:** $0 \le |A| \le 3 \times 10^4$, $|A|$ is even, $-10^5 \le A[i] \le 10^5$.

**More memory-efficient**

```python
from typing import List

class Solution:
  def canReorderDoubled(self, A: List[
    int]) -> bool
    # re-usable frequency map
    D = {}
    for x in A:
      D[x] = D.get(x, 0) + 1
    # sort keys once
    D = dict([kv for kv in sorted(list
    (D.items()),

     key=lambda x: x[0])])
    # in-place pairs by
    # updating counts
    for x in D:
      while D[x] > 0:
        D[x] -= 1
        if x <= 0:
          pair_x = x / 2
        else:
          pair_x = x * 2

        if D.get(pair_x, 0) > 0:
          D[pair_x] -= 1
        else:
          return False
    return True
```

**Less memory-efficient**

```python
from typing import List
from collections import Counter

class Solution:
  def canReorderDoubled(self, A: List[
    int]) -> bool:
    # initialize three lists (O(n))
    negs = [a for a in A if a < 0]
    pos  = [a for a in A if a > 0]
    zero = [a for a in A if a == 0]

    if any(map(lambda x: len(x) % 2 !=
      0, [negs, pos, zero])):
      return False

    if not self.is_valid(negs, True)
     or not self.is_valid(pos, False):
      return False
    return True

  def is_valid(self, A, neg=False):
    # sorted copy per bucket
    A = sorted(A)
    if neg:
      # list reverse duplicated
      A = A[::-1]
    # extra Counter (hash map)
    c = Counter(A)
    for a in A:
      if c[a] == 0:
        continue
      target = a * 2
      if c[target] == 0:
        return False
      c[a] -= 1
      c[target] -= 1
    return True
```

Figure 15: Side-by-side solutions to the problem. **(Left)** builds a single frequency map and reuses it while pairing, avoiding three full partitions (`negs`/`pos`/`zero`), extra `sorted` copies, and multiple `Counter` objects. RLM estimated memory footprint was **6518**. **(Right)** materializes three lists, sorts (and reverses) sublists, and constructs `Counter`s inside validation passes, increasing allocations and peak live objects. RLM estimated memory footprint: **10197**. (Ground truth: **6178** and **10588**, respectively.)

