# OpenReview forum: "Regression Language Models for Code"
_ICLR.cc/2026/Conference — Submitted to ICLR 2026_

### Official Review · Reviewer_NB5z · 2025-10-26

**Soundness:** 3
**Presentation:** 3
**Contribution:** 3
**Rating:** 6
**Confidence:** 2

**Summary:**

The paper proposes Regression Language Models (RLMs) that treat code-to-metric prediction as text-to-text regression with an encoder–decoder LM (initialized from T5Gemma). The same model predicts (i) peak memory for Python/C/C++ programs (APPS, CodeNet), (ii) Triton kernel latency, and (iii) NAS metrics (accuracy and multi-hardware latency) from ONNX text dumps, using a custom numeric tokenizer (P10) and constrained decoding. On APPS memory the model achieves ρ≈0.93, on CodeNet multiple languages ρ≈0.35–0.75, Triton latency ρ≈0.52, and on NAS Kendall-τ it slightly edges a strong FLAN baseline on average. The paper also shows multi-objective decoding (accuracy→latency chains), and ablations on pretraining, tokenization, head choice (decoder vs. regression head), and context length.

**Strengths:**

One small-ish (~300M) RLM covers diverse inputs (source code, ONNX graphs) and outputs (accuracy, memory, latency), reducing feature engineering and task-specific heads.

Strong APPS memory (ρ≈0.93), respectable CodeNet across 24 languages, non-trivial Triton latency, and average Kendall-τ on five NAS spaces matching/exceeding FLAN without zero-cost proxies.

**Weaknesses:**

Hardware coverage is narrow for Triton (single A6000); latency predictors can be highly hardware-sensitive. Broader devices or cross-device generalization would strengthen claims.

CodeNet regime mixes train/test questions (few-shot per problem). Authors acknowledge this limits zero-shot difficulty; results may over-estimate real-world generalization to new problems with unseen inputs.

**Questions:**

How many samples per input are used at inference, and how are means/medians chosen? Did you evaluate calibration or UQ quality (CRPS/NLL) when using the density view?

The ONNX graphs can be long; what fraction of graphs truncate at 1k/2k/4k tokens, and how does that affect τ/ρ?

---

> ### Author Response · Authors · 2025-11-15
>
> Hi Reviewer NB5z, thank you for your thorough and positive review! Please find our responses below:
>
> > Hardware coverage is narrow for Triton (single A6000)
>
> We believe showing the RLM also works when conditioned on the hardware / GPU spec is a great opportunity, but for another paper due to its larger scope and dataset requirements. We emphasize that we do have high hardware coverage for latency prediction already, across 4 distinct devices (Nvidia GPU, Intel CPU, Mobile (Pixel 3), ASIC (Eyeriss)) in our multi-objective results in Section 5.
>
> > CodeNet regime mixes train/test questions (few-shot per problem). Authors acknowledge this limits zero-shot difficulty;
>
> Good catch: We agree that this was an issue with the CodeNets dataset, and we raised this issue in Section 4.1, but also mentioned that this also tests the RLM on limited information and few-shot scenarios. For example, we do zero-shot (limited information) tests on new languages in Table 10, where problems may be repeated but in entirely new (unseen) languages, where RLM exhibits non-trivial performance. Note that this issue doesn’t exist for Triton kernels, as well as all of our NAS results which match SoTA hand-crafted GNN baselines.
>
> > How many samples per input … Did you evaluate calibration or UQ quality (CRPS/NLL) when using the density view?
>
> | Metric        | Mean CRPS | Mean NLL |
> |---------------|----------:|---------:|
> | Accuracy |   0.0817  |   0.5224 |
> | Latency Pixel3   (Mobile) |  0.0678  |    -0.6558 |
> | Latency Eyeriss (ASIC)  |   0.1076  |  1.5002 |
> | Latency gold_6226 (CPU) |   0.0362  |  -1.3033 |
> | Latency titanx_1  (GPU) |   0.0407  |  -1.1157 |
>
>
> In Figure 5, we use 64 samples per input, and take the median of the samples on a per-objective basis for the per-device correlation calculation. Across four metrics we obtain low CRPS (~0.04–0.08) and modest or negative NLL, indicating well calibrated predictive distributions.
>   - We note that Eyeriss (ASIC) latencies are heavily discretized: many different architectures end up with exactly the same runtime (the vertical bands in Fig. 5). This hardware effect makes the latency distribution highly discrete, so a Gaussian fit naturally gives a worse NLL on this metric even though the underlying point predictions are still reasonable; in other words, this behaviour is a property of the device rather than a failure of the model.
>
> > The ONNX graphs can be long; what fraction of graphs truncate at 1k/2k/4k tokens
>
> Good question. We did have to truncate all of the ONNX graphs, but note that typically NAS graphs use repeated designs stacked across each other, and so truncation would only eliminate duplicate information, which should not harm performance. Furthermore, with a larger compute budget or efficient attention mechanisms, one can always scale up the sequence length limit of the model to fully process the graph.

---

> > ### Author Response · Authors · 2025-11-25
> > **Kind follow-up on our rebuttal**
> >
> > Dear Reviewer NB5z,
> >
> > We hope you are doing well. Just a short note to inform that our rebuttal now:
> > - Clarifies that our study not only targets Triton with A6000, but multi-objective NAS experiments already cover four heterogeneous devices (mobile, CPU, GPU and ASIC).
> > - Details our probabilistic evaluation using 64 samples per input with per-objective medians, and we report the CRPS and NLL across 4 device-specific latencies and accuracy, showing reasonably calibrated predictive distributions.
> >
> > Thank you again for your positive review, we appreciate your time and would be grateful if you could consider these clarifications in your final assessment.
> >
> > Regards,
> >
> > Tha Authors

---

> > > ### Comment · Reviewer_NB5z · 2025-11-26
> > >
> > > I especially appreciate the additional explanations the calibration metrics (CRPS/NLL) with sampling. However, I am not familar with this specific sub-area, so I will maintain my original score, and my confidence in the evaluation remains low.

---

### Official Review · Reviewer_sjYR · 2025-10-27

**Soundness:** 3
**Presentation:** 3
**Contribution:** 3
**Rating:** 4
**Confidence:** 3

**Summary:**

This paper introduces Regression Language Models that frame code-to-metric regression as a text-to-text generation task. Instead of relying on domain-specific feature engineering, the authors propose using a unified encoder-decoder model that reads the raw text of code or intermediate representations and autoregressively decodes the numerical metric using a special P10 tokenizer.

**Strengths:**

1. The paper's core idea, i.e., treating all code regression as a text-generation problem, is original and simplifies a process that traditionally requires heavy, domain-specific feature engineering.
2. A key strength is the demonstration of a unified model successfully handling regression tasks across highly disparate inputs and predicting different metrics.
3. The methods outperform or remain competitive with baselines.

**Weaknesses:**

1. The paper's central choice to formulate regression as an autoregressive token-generation problem (using P10) instead of using a standard regression head is not convincingly justified. Have you tried using a log transform for the output within the MLP? It's also not clear if the conditional (auto-regressive) modeling provides any real benefit over a simpler multi-head approach that predicts all metrics independently from the encoder embedding.
2. The paper motivates its approach by contrasting it with feature-engineering methods. However, it only compares against deep learning feature engineering. For tasks like memory and latency prediction, the true established baselines are often traditional static analysis tools or logic-based methods from the programming languages and compiler communities. The paper provides no comparison to these methods.
3. The study relies exclusively on T5Gemma. This is not a common base model in modern LLM research (compared to Llama, Qwen, etc.).
- The authors should include results based on more standard base models.
- The paper should also report the performance of the base T5Gemma model without RLM fine-tuning to properly isolate the gains from the proposed training method.
4. Since the task is reformulated as text-to-text generation, the most obvious baselines are missing: the performance of modern, general-purpose LLMs. How well do state-of-the-art models (e.g., GPT, Gemini, DeepSeek) perform on this task under different settings? This comparison is essential to understand if the specialized RLM is necessary or if the capability is already present in general-purpose models.

**Questions:**

See Weaknesses.

---

> ### Author Response · Authors · 2025-11-15
>
> Hi Reviewer sjYR - thanks for your detailed review of our paper. Please find our responses below, especially with additional experiments as requested.
> > The paper's central choice to formulate regression as an autoregressive token-generation problem (using P10) instead of using a standard regression head is not convincingly justified. Have you tried using a log transform for the output within the MLP?
>
> We understand that using a decoder head is controversial and not fully mainstream (time will tell), but it has numerous benefits if done right:
> - No need for normalizations or transformations if using a token representation which accounts for exponents.
> - Stable multi-task training, due to the cross-entropy loss for each example being equally weighted.
> - Universal approximation over any density p(y|x).
>
> We further refer the reader to previous works also which use some variant of decoding / categorical tokenization of numbers:
> - [1] which establishes in controlled, isolated settings, decoder heads are just as precise as MLP heads.
> - [2] which is used for tabular regression
> - [3] which is used for distributional reinforcement learning.
>
> Regarding the log-transform output within the MLP, we match the settings described in Table 5, and train a LogNorm Regression Head; it performs worse than Normalized Regression head, which 0-1 normalizes each domain in the dataset. We can indeed keep engineering the output scaling of our data, but a key feature of the RLM is that it removes this exact form of feature engineering.
>
> | Head | Spearman-Rho |
> |----------------|-------|
> | Regression Head | 0.478 |
> | LogNorm Regression Head | 0.512 |
> | Normalized Regression Head | 0.717 |
> | Decoder Head (Ours) | **0.800** |
>
>
> > simpler multi-head approach that predicts all metrics independently from the encoder embedding.
>
> - Embedding + MLP head: A core issue of embedding methods is their compressive nature, as the full token output of the Transformer needs to be pooled into a fixed length vector. This leads to suboptimal performance on the single-objective case, as shown in the table above.
> - Multi-objective correlated modeling: Note that the decoder head’s universal density approximation ability also extends to multi-objective settings to capture arbitrary correlations between different metrics. No other regression method is capable of this - even Gaussian Processes are only able to model Gaussian covariance matrices (which cannot model e.g. multi-modal distributions, common in real world regression tasks). We also refer to numerous previous works which establish the benefit of correlated predictions [4,5,6,7,8].
>
> > For tasks like memory and latency prediction, the true established baselines are often traditional static analysis tools or logic-based methods from the programming languages and compiler communities.
>
> We are aware of the various types of static analysis tools from other communities, and we have done our due diligence to cite related works out of academic respect in our paper. However, the regime we target is much broader, APPS and CodeNet contain arbitrary competitive programming submissions with dynamic allocation, recursion, container libraries etc., and our NAS experiments operate on large ONNX graphs.
>
> - Existing analyzers in PL and architecture assume a fixed IR plus a hand-crafted model of per-operation cost with components for caches, occupancy and pipeline stalls, overall only accurate when compiler and hardware are carefully modeled, and do not scale to predicting numeric peak memory or latency for arbitrary programs across the breadth of our datasets.
> - In contrast, RLMs learn directly from measurements and can be applied unchanged across languages! Note that in Table 10, we find that the RLM exhibits non-trivial rank performance on languages that it has never been trained on. Building static tools with comparable coverage would be a large systems effort and outside the scope of this work as a baseline.
> - We will expand the related work to position classic static methods as complementary for deterministic, tightly controlled code, whereas resolving the issue of predicting on noisy, real executions without any task-specific feature engineering or hardware modelling is a core contribution of our paper.

---

> > ### Author Response · Authors · 2025-11-15
> > **continuation of response to reviewer sjYR**
> >
> > > The study relies exclusively on T5Gemma
> >
> > Due to the large compute resources required, we can always append more results with open-source LLMs after the rebuttal phase. But please keep in mind, selecting the right pretrained LLM isn’t our message here (this is effectively a hyperparameter tuning exercise).  T5Gemma is only used as a convenient pretrained model to show that pretrained models can help code prediction over randomly initialized ones.
> >
> >
> > > “The paper should also report the performance of the base T5Gemma model without RLM fine-tuning” and “Since the task is reformulated as text-to-text generation, the most obvious baselines are missing: the performance of modern, general-purpose LLMs.”
> >
> > For T5Gemma, the base model **completely fails to predict the regression target** with raw ICL. This is because of poor instruction following as well as lack of structured output support. We present a more appropriate baseline, where we use the GPT-5-Thinking model (see results shown in [above](https://openreview.net/forum?id=KnzEMXSK3f&noteId=vNWKu8m6vP)). Even with “MAX-Shot ICL” (maximum possible in-context examples) to GPT5, it is significantly worse than RLMs. Furthermore, the cost to evaluate Max-Shot ICL was higher than the entire training cost of the RLM.
> >
> >
> > [1] Song, X., & Bahri, D. (2025). Decoding-based Regression. arXiv preprint arXiv:2501.19383.
> >
> > [2] Hollmann, N., Müller, S., Purucker, L., Krishnakumar, A., Körfer, M., Hoo, S. B., ... & Hutter, F. (2025). Accurate predictions on small data with a tabular foundation model. Nature, 637(8045), 319-326.
> >
> > [3] Bellemare, M. G., Dabney, W., & Munos, R. (2017, July). A distributional perspective on reinforcement learning. In International conference on machine learning (pp. 449-458). PMLR.
> >
> > [4] Bonilla, E. V., Chai, K., & Williams, C. (2007). Multi-task Gaussian process prediction. Advances in neural information processing systems, 20.
> >
> > [5] Pearce, M. A. L. (2019). Bayesian optimisation with multi-task Gaussian processes (Doctoral dissertation, University of Warwick).
> >
> > [6] Palar, P. S., Zuhal, L. R., Chugh, T., & Rahat, A. (2020). On the impact of covariance functions in multi-objective Bayesian optimization for engineering design. In AIAA Scitech 2020 Forum (p. 1867).
> >
> > [7] Kaifeng Yang, Kai Chen, Michael Affenzeller, and Bernhard Werth. 2023. A New Acquisition Function for Multi-objective Bayesian Optimization: Correlated Probability of Improvement. In Proceedings of the Companion Conference on Genetic and Evolutionary Computation
> >
> > [8] Swersky, K., Snoek, J., & Adams, R. P. (2013). Multi-task bayesian optimization. Advances in neural information processing systems, 26.

---

> ### Comment · Reviewer_sjYR · 2025-11-17
>
> Thank you for your timely and detailed response!
>
> I find it quite surprising that a model with strong instruction-following capabilities, like the 'GPT-5-Thinking' you mentioned, would perform so poorly on these tasks. Could you please provide some specific examples of its failures, along with an error analysis? Understanding how it fails would be very insightful.
>
> Regarding the base model, I understand the constraints on computing resources. However, the concern remains that T5Gemma is not a standard or widely used baseline in current LLM research. To make your results more grounded and comparable to other work in the field, would it be possible to provide results using a more conventional model? For instance, the Qwen3 series also offers small-scale models, such as a 0.6B variant, which would serve as a more widely recognized baseline.

---

> > ### Author Response · Authors · 2025-11-18
> > **GPT5 regression output analysis, Qwen and Llama regressors**
> >
> > Thank you for your response! We studied the regression outputs from our GPT-5-Thinking ablation, as well as trained the requested models, with explanations below. We hope this helps clarify why encoder-decoder models with specialized vocabulary are effective.
> >
> >
> > > I find it quite surprising that a model with strong instruction-following capabilities, like the 'GPT-5-Thinking' you mentioned, would perform so poorly on these tasks. Could you please provide some specific examples of its failures, along with an error analysis? Understanding how it fails would be very insightful.
> >
> >
> > GPT-5 unfortunately does not provide its chains of thought. But note that regression is inherently an inductive (i.e. intuitive / unexplainable by words), not deductive reasoning problem anyways, and thus chains of thought with natural language may not even be very useful for understanding behavior. Thus, we find that ‘outputting extremely precise numbers’ with no fine-tuning is a very difficult task, and is not entirely surprising that GPT-5 does not have the fidelity to regress accurately on large code/graph samples.
> >
> >
> > Instead, we analyzed its behavior numerically on APPS dataset - we found that most of the regression predictions from MAX-Shot ICL have y-values clustered within the ranges (73-78) and (94-100), the model therefore tends to cluster its predictions into those ranges. But within these clusters, GPT5 is unable to distinguish the worst and best examples with high precision - it obtains within-cluster Spearman ranks of 0.2018 and 0.3755 respectively, while the RLM obtains within-cluster Spearman ranks of  0.5544 and 0.8106, respectively.
> >
> > > Regarding the base model, I understand the constraints on computing resources. However, the concern remains that T5Gemma is not a standard or widely used baseline in current LLM research. To make your results more grounded and comparable to other work in the field, would it be possible to provide results using a more conventional model? For instance, the Qwen3 series also offers small-scale models, such as a 0.6B variant, which would serve as a more widely recognized baseline.
> >
> >
> > We use T5Gemma because it is a frontier, very recent encoder-decoder model. The model class even outperforms Gemma 2 at the 2B and 9B model sizes on tasks such as GSM8K, MMLU etc. Further, from Figure 3 of the T5Gemma paper [1],  it achieves a better pareto-front then decoder-only models across all ranges of inference flop budgets. As such, it is the frontier, conventional encoder decoder model of choice that should serve as a recognized baseline.
> >
> >
> > On our specific design with the custom P10 decoder: note that using the original model’s tokenization to decode numbers is significantly worse (Figure 9, P10 delivers a rank-correlation of 0.747, whereas using T5Gemma default vocabulary achieves 0.654). This is because without the digit vocabulary, the model has to learn to map its entire default vocabulary to digits. P10 tokenization (which gives the most significant boost according to our findings) is **not possible for conventional, decoder-only models**, as the decoded digits need to have cross-attention with the input sequence as well as be auto-regressive (and thus, the vocabulary has to be the same).
> >
> >
> > To further justify our design choice within the discussion deadline, we adapt our code-base to train three models:
> >
> > - Our default T5Gemma-s-s-prefixlm RLM with P10 tokenizer
> > - Decoder-only model Qwen/Qwen3-0.6B
> > - Decoder-only model Meta-Llama-3.2-1B.
> >
> >
> > For Qwen and Llama, we use their default vocabulary and freeze the first half of the decoder layers (to match the strategy of freezing the encoder layers from T5Gemma). We train with 128K examples from CodeNets, 80K examples from APPS and 10K examples from KernelBook. The sequence length is 1024. We evaluate the trained models on 512 examples from APPS, and choose the median from 128 samples per input.
> >
> > Note that Qwen and Llama often struggle to decode valid digits due to their unconstrained natural language vocabulary.  The findings below are consistent with our expectation that there is a major advantage to using encoder-decoder models, with a specialized decoder vocabulary. With extensive training and larger datasets, it may be possible to learn to map the natural-language focused vocabulary to digit decoding, but is extremely sample-inefficient.
> >
> > | Model | APPS Spearman-Rho |
> > |-|-|
> > | T5Gemma-s-s-prefixlm (P10 Vocab) | **0.8982** |
> > | Llama-3.2-1B (Default Vocab) | 0.1715 |
> > | Qwen3-0.6B (Default Vocab) | 0.1453 |
> >
> >
> > [1] Zhang, B., Moiseev, F., Ainslie, J., Suganthan, P., Ma, M., Bhupatiraju, S., ... & Dong, Z. (2025). Encoder-Decoder Gemma: Improving the Quality-Efficiency Trade-Off via Adaptation. arXiv preprint arXiv:2504.06225.

---

> > > ### Author Response · Authors · 2025-11-25
> > > **Kind follow-up on our rebuttal**
> > >
> > > Dear Reviewer sjYR,
> > >
> > > We hope you are doing well, following your questions, our rebuttal now:
> > >
> > > - includes an explicit ablation comparing several MLP heads (log-transform head, normalized regression head) against our P10 decoder head, showing that decoder head achieves the strongest spearman correlation and supports multi-objective regression.
> > > - Adds large model and alternative backbone baselines, including MAX-Shot GPT-5-Thinking ICL, and small decoder-only regressors based on Qwen3-0.6B and Llama-3.2-1B, which underperform the specialised RLM.
> > > - Motivates our choice for T5Gemma and custom numeric vocabulary, with demonstrations that using default vocabulary significantly degrades rank correlation.
> > >
> > > We would be grateful if you could take a quick look at these additions and clarifications, and let us know of any remaining concerns.
> > >
> > > Regards,
> > >
> > > The Authors

---

> ### Comment · Reviewer_sjYR · 2025-11-27
>
> Thank you for the reply and I have increased my rating.

---

> > ### Author Response · Authors · 2025-11-28
> > **Thank you!**
> >
> > Thank you very much and for your comprehensive review!

---

### Official Review · Reviewer_e7RV · 2025-11-01

**Soundness:** 3
**Presentation:** 3
**Contribution:** 2
**Rating:** 4
**Confidence:** 2

**Summary:**

The paper proposes Regression Language Models for code. It introduces encoder-decoder language models based on T5Gemma, fine-tuned to perform numeric regression on code-related tasks. They have discussed predicting memory footprint, latency, or model accuracy from textual code or ONNX representations. The authors claim that a single unified model can regress across diverse programming languages and the created NAS spaces, outperforming specialized graph-based regression models (e.g., GNNs, FLAN).

**Strengths:**

1. It is a very interesting direction for code applications: predicting performance metrics directly from code.
2. The idea of using a unified single model across code, kernel, and graph inputs is conceptually appealing.
3. The presentation of the paper is organized with supportive tables and illustrative examples.

**Weaknesses:**

1. Novelty: I can hardly agree that the major concept is new. It uses text-to-text regression and autoregressive regression with trivial modifications.
2. Missing details of the experiments.
a. The multi-task regression setup mixes heterogeneous tasks without normalization or loss balancing.
b. No ablation on negative transfer or cross-domain interference.
The results thus cannot support the central claim that “a unified RLM generalizes across tasks.”
3. Evaluation
a. The >0.9 Spearman on APPS is due to the dataset’s extremely small label variance.
b. The gains in table 4 are statistically insignificant.
4. In general, the authors are using inflated language such as “massively simplifying graph regression” and "general-purpose universal regressor,” with no mechanistic or theoretical support.

**Questions:**

1. Appendix A.3 shows that fine-tuning reduces performance due to catastrophic forgetting, and A.2 shows that multilingual pretraining hurts zero-shot languages. Do they directly contradict the paper’s claims of generality?
2. How does the model handle tasks with vastly different output scales (e.g., bytes vs milliseconds vs accuracy)?
3.Why compare to FLAN without describing its training details?

---

> ### Author Response · Authors · 2025-11-15
>
> Hi Reviewer e7RV, thanks for the detailed review of our paper. We added clarifications on our contributions below:
>
>
> > Novelty: I can hardly agree that the major concept is new. It uses text-to-text regression and autoregressive regression with trivial modifications.
>
> The novelty isn’t coming from the method (it’s simple by intention). The novelty is coming from the results - for example it is very surprising one can achieve *SOTA* on NAS graphs, without any handcrafted features. The vision is being able to eliminate feature engineering altogether, even for code-related problems.
>
> > mixes heterogeneous tasks without normalization or loss balancing.
>
> This is intentional and a key feature of our work - because our loss is cross-entropy over text tokens, it’s scale-invariant and doesn’t need normalizations over multiple tasks, allowing easy multi-task training.
>
> > “No ablation on negative transfer or cross-domain interference”
>
> We explicitly checked for negative transfer in Appendix A. A single-task RLM trained only on code achieves Spearman rank = 0.701, while the unified RLM (Code+Graph model in Figure 10, Table 9) achieves Spearman rank = 0.706. This suggests that adding other domains does not introduce noticeable negative transfer. This is especially a strong case, as code  regression and NAS graph regression are very different tasks (former is predicting latency / memory, latter is predicting classification accuracy).
>
> > The >0.9 Spearman on APPS is due to the dataset’s extremely small label variance
>
> “Small label variance and spearman rank” - Can the reviewer clarify what they mean by “small label variance”? The y-values of APPs (Figure 2) range from 10^3 to 10^6, which is very wide actually. But also, Spearman rank is independent of y-value scaling anyways, making it a clean signal of regression performance.
>
> > The gains in table 4 are statistically insignificant.
>
> This is a very important point, note that the results in Table 4 are for ‘graph regression’. On one hand, we have completely generic ONNX code input to our RLM for several NAS spaces, whereas FLAN uses a GNN on a hand-crafted description of each search space. These graphs are huge, as shown in Appendix D.6. The regression task is to predict the classification accuracy the graph would achieve, if trained on CIFAR-10. We compare with GNNs, as well as FLAN, a hand-crafted solution that uses a specialized graph neural network (combining graph-attention, symmetry breaking techniques specialized for NAS, and forward-backward passing on the DAG etc. [1, 2]), the key contribution in Table 4 is to demonstrate that RLMs require zero feature engineering, directly ingest the entire ONNX graph of these architectures, and can actually **match** the performance of FLAN, a highly specialized regressor for NAS. In summary, significant efforts in prior work were to find better features to perform prediction, our work circumvents the whole process by having completely generic regression over ONNX graphs.
>
> > authors are using inflated language
>
> Thank you for this note, we can tone down this inflated language. But to provide some context for our phrasing:
> - “Massively simplifying graph regression”: In the NAS community, there is an ocean of work (referenced in our paper) specifically addressing regression for neural network graph architectures. A major problem is feature engineering (complex encodings, graph kernels, etc.) which are also non-transferrable. Simply using ONNX as a unified representation is significantly easier.
> - “General purpose universal regressor”: The text-based representation for both inputs and outputs allows large-scale regression pretraining at ease. Since language models are Turing machines [3], they are also able to learn any density p(y | x) which can be approximated by a computational process, which is a fundamental approximation guarantee.
>
>
> > Appendix A.3 shows that fine-tuning reduces performance due to catastrophic forgetting
>
> Yes, this may occur after fine-tuning which specializes the model to a domain. But this is a general behavior of LLMs, not specific to the RLM, and can be mitigated with larger models, LoRA, etc.
>
> > A.2 shows that multilingual pretraining hurts zero-shot languages. Do they directly contradict the paper’s claims of generality?
>
> They do not, as discussed in A.2, CodeNet only have a limited set of distinct problems, in practice, seeing more diverse problems in a single language may be more helpful than seeing the same problem repeated across multiple languages.

---

> > ### Author Response · Authors · 2025-11-15
> > **continuation of response to reviewer e7RV**
> >
> > > How does the model handle tasks with vastly different output scales (e.g., bytes vs milliseconds vs accuracy)?
> >
> > As mentioned in our response for “Weakness 2”, because the loss is cross-entropy over floating point tokenizations which cover the real number line, it is robust to differences in output scales, and requires no y-normalizations, in comparison to an MLP head. Despite the discrete nature of tokenization, it has been established [4] that decoding numeric tokens can be just as accurate as a MLP head.
> >
> >
> > > 3.Why compare to FLAN without describing its training details?
> >
> > FLAN [2] is a SoTA NAS graph-regression baseline. We refer the reader to that paper, and we will make sure to add more  details of that baseline in our paper.
> >
> > **References**
> >
> > [1] Ning, X., Zhou, Z., Zhao, J., Zhao, T., Deng, Y., Tang, C., ... & Wang, Y. (2022). TA-GATES: An encoding scheme for neural network architectures. Advances in Neural Information Processing Systems, 35, 32325-32339.
> >
> > [2] Akhauri, Y., & Abdelfattah, M. S. (2024). Encodings for prediction-based neural architecture search. arXiv preprint arXiv:2403.02484.
> >
> > [3] Pérez, J., Barceló, P., & Marinkovic, J. (2021). Attention is turing-complete. Journal of Machine Learning Research, 22(75), 1-35.
> >
> > [4] Song, X., & Bahri, D. (2025). Decoding-based Regression. arXiv preprint arXiv:2501.19383.

---

> > > ### Author Response · Authors · 2025-11-25
> > > **Kind follow-up on our rebuttal**
> > >
> > > Dear Reviewer e7RV,
> > >
> > > We hope you are doing well. We wanted to kindly let you know that our rebuttal now:
> > > - Clarifies that the main novelty lies in eliminating feature engineering while matching or surpassing state-of-the-art graph-based baselines on NAS, as well as code regression.
> > > - Explains why our multi-task setup does not require loss balancing or normalization (scale-invariant cross-entropy over numeric tokens). We highlight the single-task vs. unified RLM results, showing no noticeable negative transfer between code and NAS Graph Regression.
> > >
> > > If you have a chance to revisit the paper with these clarifications, we would greatly appreciate further discussion.
> > >
> > > Regards,
> > >
> > > The Authors

---

> > ### Comment · Reviewer_e7RV · 2025-11-26
> >
> > Thank you to the authors for the detailed response.
> >
> > > The novelty isn’t coming from the method (it’s simple by intention). The novelty is coming from the results - for example it is very surprising one can achieve SOTA on NAS graphs, without any handcrafted features. The vision is being able to eliminate feature engineering altogether, even for code-related problems.
> >
> > While the empirical results are promising, in the context of ICLR expectations, a paper must demonstrate methodological or conceptual advancement rather than relying solely on empirical surprise. Prior NAS works have already moved beyond handcrafted features by using GNN-based architecture encoders and transformer-based graph embeddings, so eliminating manual feature design is not a new capability unique to this work.
> >
> > More importantly, the paper does not provide mechanistic analysis explaining how the model extracts meaningful structural or semantic signal from ONNX text, nor evidence that it learns execution-relevant properties rather than shallow token-level correlations. Without such analysis, the observed performance remains an unexplained empirical outcome rather than a demonstrated general capability.
> >
> > Furthermore, a comparison with real profiling tools would be essential to contextualize the predictive performance of this method and to evaluate whether it offers advantages over conventional direct measurement approaches, which already provide precise ground-truth measurements.
> >
> > > This is intentional and a key feature of our work - because our loss is cross-entropy over text tokens, it’s scale-invariant and doesn’t need normalizations over multiple tasks, allowing easy multi-task training.
> >
> > In my understanding, even with CE, tasks with different distributional properties (e.g., sequence length, operator diversity, entropy of token transitions) can induce unequal gradient magnitudes, leading to implicit over- or under-training of certain tasks. Without empirical balancing or analysis of inter-task interference, the assumption of “no need for normalization” remains unsubstantiated.
> >
> > > We explicitly checked for negative transfer in Appendix A. A single-task RLM trained only on code achieves Spearman rank = 0.701, while the unified RLM (Code+Graph model in Figure 10, Table 9) achieves Spearman rank = 0.706. This suggests that adding other domains does not introduce noticeable negative transfer. This is especially a strong case, as code regression and NAS graph regression are very different tasks (former is predicting latency / memory, latter is predicting classification accuracy).
> >
> > A difference of 0.005 in Spearman rank cannot be meaningfully interpreted without variance estimates, or statistical testing. Given typical noise levels in regression evaluation, this difference is well within expected fluctuation and does not constitute evidence against negative transfer.
> >
> > > “Small label variance and spearman rank” - Can the reviewer clarify what they mean by “small label variance”? The y-values of APPs (Figure 2) range from 10^3 to 10^6, which is very wide actually. But also, Spearman rank is independent of y-value scaling anyways, making it a clean signal of regression performance.
> >
> > I am referring to within-problem variance, not global variance across all programs. For many APPS problems, the different code solutions for the same task exhibit extremely similar runtime or memory measurements, leading to very limited spread in labels within each equivalence class.
> >
> >
> > > The gains in table 4 are statistically insignificant.
> >
> > The rebuttal emphasizes the convenience of a text-based ONNX representation and the perceived elimination of feature engineering, but it does not address the original statistical concern. Matching FLAN’s performance is interesting, but without reporting variance across runs, confidence intervals, seed sensitivity, or robustness metrics, it is not statistically valid to claim that the observed~0.002 difference constitutes equivalence in capability.

---

> > > ### Author Response · Authors · 2025-11-28
> > >
> > > Thank you for your thorough review and feedback, please find our responses below:
> > >
> > > > a paper must demonstrate methodological or conceptual advancement rather than relying solely on empirical surprise.
> > >
> > > We believe our paper is in fact a methodological and conceptual advancement, as it demonstrates the strength and simplicity of using pretrained encoder-decoders, bolstered by clear ablations in Section 6 to show why the RLM works. This also includes our response to Reviewer sjYR, showing the failure of GPT5-like models to do ICL on complex regression tasks, as well as failure of decoder-only fine-tuned models. Our release of all datasets will also allow further competitive benchmarking which enables further not just on RLMs, but also GNNs or static code analysis.
> > >
> > > We’d like to further point out the [ICLR 2026 Reviewer Guide](https://iclr.cc/Conferences/2026/ReviewerGuide), which says a submission has value when it “demonstrates new, relevant, impactful knowledge (incl., empirical, theoretical, for practitioners, etc)” which we satisfy.
> > >
> > > > Prior NAS works have already moved beyond handcrafted features by using GNN-based architecture encoders and transformer-based graph embedding
> > >
> > > Our disagreement seems to be on the subjective definition of “handcrafted” - we take it to mean methods which lack flexibility and robustness to new tasks, and will make this more explicit in the paper. GNNs and Transformer-based graph embeddings (e.g. in FLAN) still require strict, fixed-dimensional encoding of operation matrices and are not robust to new operations. Any single change to any of these operations requires retraining of a whole autoencoder (Arch2Vec) or retraining of CATE-style encoders.
> > >
> > > Furthermore, note that our results are compared to “FLAN-ZCP” which doesn’t rely on GNN (Arch2Vec) or Transformer-based graph embeddings (CATE), but instead relies on **13 handcrafted zero-cost proxies** adapted from prior work. Our method significantly outperforms FLAN, CATE, FLAN Arch2Vec variants and is on-par with FLAN-ZCP, which requires real, significant engineering time to handcraft ZCP features. ZCP generation for training the base predictor in FLAN required 6.6 million evaluations of architectures across 18 NAS Tasks; **our RLM requires none of that**, working simply on ONNX graphs.
> > >
> > > > mechanistic analysis explaining how the model extracts meaningful structural or semantic signal from ONNX text,
> > >
> > >
> > > General regression is inherently a non-semantic and non-deductive reasoning task for which there does not always exist explicit interpretable reasoning, which also applies to classic regression models such as MLPs and random forests. The most standard way to perform regression interpretability is to perform Shapley regression (i.e. change an input feature slightly and see how the output changes too), but “small feature changes” are not well-defined for combinatorial and code spaces. Nonetheless, we affirm that Figure 4 and Appendix D7 all demonstrate such changes in model prediction when comparing minor syntax changes in submissions to the same question.
> > >
> > > If the reviewer is referring to mechanistic interpretability, we believe this is a relatively new field with no clear guarantees, and is largely out of the scope of this paper.
> > >
> > > > nor evidence that it learns execution-relevant properties rather than shallow token-level correlations.
> > >
> > > Note that [1] attention is Turing-complete, and theoretically language models can already imitate Turing machines (i.e. code executions). As for whether the RLM is learning “shallow token-level correlations” - we believe this debate is completely out of the scope of this paper, given the enormous broader discussions surrounding whether LLMs can or cannot reason [2], or are merely e.g.. “stochastic parrots” [3].
> > >
> > > In any case, we’ve shown that over test examples, the RLM achieves good regression results, as a simple and clean evidence of its ability.
> > >
> > > [1] https://jmlr.org/papers/v22/20-302.html
> > >
> > > [2] https://machinelearning.apple.com/research/illusion-of-thinking
> > >
> > > [3] https://dl.acm.org/doi/10.1145/3442188.3445922
> > >
> > >
> > > > comparison with real profiling tools would be essential to contextualize the predictive performance of this method
> > >
> > > If the reviewer could kindly provide **a single open-source framework that works on the breadth of code we discussed** for a fair comparison, we would be grateful to run it and will try our best to get it before the rebuttal deadline. However, the only reliable framework we found was emmtrix, which is a ‘closed source demo’ which is not available to us, further it is a static analysis tool, not a profiler. Unfortunately, while real profiling tools sound useful, we are simply unable to find a comparable option and would appreciate the help!

---

> > > > ### Author Response · Authors · 2025-11-28
> > > >
> > > > > CE, tasks with different distributional properties (e.g., sequence length, operator diversity, entropy of token transitions) can induce unequal gradient magnitudes, leading to implicit over- or under-training of certain tasks.
> > > >
> > > > Thank you for this note - we agree and we will adjust our wording! Our intention was to say that CE loss removes the need for output scale normalization (bytes, milliseconds, accuracies), which is a major practical advantage over MSE heads. Regarding task imbalance, we do not claim any theoretical guarantee, we provide empirical evidence that the unified RLM matches or improves single-task baselines on code and NAS benchmarks, and we do not observe any obvious negative transfer.
> > > >
> > > >
> > > > > A difference of 0.005 in Spearman rank cannot be meaningfully interpreted without variance estimates, or statistical testing.
> > > >
> > > > Please note that the difference of 0.005 is the **aggregate** of evaluation over APPS (Python Memory Overhead), KernelBook (Triton Kernel Latency Estimation), and CodeNets (three languages, C, C++ and Python). Unfortunately, it is outside the scope of the rebuttal period to train 5 regression language models across two settings, but we hope you understand that Table 9 is not an isolated number, but is an aggregate of 5 very distinct test-settings, which significantly reduces the statistical variance.
> > > >
> > > > Furthermore, Figure 10 shows the train-time Spearman-rank correlation on Triton Kernel Latency Estimation, which has more than 50 measurements throughout the training process, and we can clearly see that both variants of the model follow very closely, with no evidence of negative transfer.
> > > >
> > > > > I am referring to within-problem variance, not global variance across all programs.
> > > >
> > > > Note that lower within-problem variance would actually make the task of discriminating between solutions **more difficult** when the metric is Spearman rank. In this paper, we profiled each solution ourselves by measuring the within-problem variance of memory across the APPS dataset, and found huge variance in the within-problem solutions.
> > > >
> > > > Suppose we isolated problems with at least 4 solutions,  yielding 3900 problems.
> > > >
> > > > - Across this, the median per-problem standard deviation is 450.8, with a mean of 56000.
> > > > - The 90th percentile per-problem std-dev is 35000!
> > > > - There is a heavy tail of problems where different solutions give a wide range of memory overheads. If we use the median, 40% of problems have within-problem standard deviation more than 2 times the median! 35% exceed 3 times the median.
> > > >
> > > > Thus, there is **very substantial within-problem variability** in the target across solutions and find that the task is not limited by lack of signal, and even within-problems, our results are quite significant.
> > > >
> > > > > Matching FLAN’s performance is interesting, but without reporting variance across runs, confidence intervals, seed sensitivity, or robustness metrics, it is not statistically valid to claim that the observed~0.002 difference constitutes equivalence in capability.
> > > >
> > > > Thank you for this note. Please note, as was true for our test on Code and Code+Graphs, this one is also an **aggregate** of 5 fine-tuning runs on entirely different NAS spaces. We find that the training and fine-tuning runs are fairly stable - to provide more evidence of this within the discussion deadline, we fine-tune `gemma-s-s-prefixlm` models with 5 independent seeds on the NAS dataset, for the DARTS baseline from Table 4 in identical settings, and find the following kendall-tau correlation across the runs:  (0.54446, 0.56951, 0.53327, 0.55142, 0.52800). Our reported result in Table 4 is lower than the other runs reported here, with different seeds.
> > > >
> > > > We sincerely hope our clarifications on contribution guidelines, static analysis tools, APPS problem variance analysis and repetition of the RLM baseline on DARTS help demonstrate the value regression language models have.

---

### Official Review · Reviewer_AHw1 · 2025-11-01

**Soundness:** 2
**Presentation:** 2
**Contribution:** 2
**Rating:** 4
**Confidence:** 2

**Summary:**

This paper introduces a unified framework called Regression Language Models (RLM) for "code-to-metric" regression. The core idea is to predict various numerical metrics directly from source code and other textual representations of programs, eliminating the need for manual feature engineering.

**Strengths:**

- Elimination of Feature Engineering: A significant advantage of the RLM is its ability to learn directly from raw text representations of code. This simplifies the prediction pipeline and makes it more adaptable to new programming languages, hardware, or model architectures.
- Strong Empirical Results: The RLM achieves impressive performance, with a Spearman's rank correlation greater than 0.9 on predicting the memory usage of competitive programming submissions. It also performs well on predicting Triton kernel latency and achieves state-of-the-art results in Neural Architecture Search benchmarks, outperforming graph-based methods.[1][2][3][4]
Multi-Objective Prediction: The autoregressive nature of the decoder allows for the conditional prediction of multiple metrics. This is a powerful feature for tasks like hardware co-design and compiler optimization, where trade-offs between different performance aspects need to be considered.
- Effective Use of Pretraining: The work demonstrates the significant benefit of initializing the RLM from a pretrained language model (T5Gemma), which leads to faster convergence and better overall performance.

**Weaknesses:**

- Limited Exploration of Larger Models: While the paper shows promising results with a 300M and a 600M parameter model, a more extensive analysis of how performance scales with model size would be beneficial. A marjor concern is that: modern SOTA models excel at generalizing from just a handful of examples provided in a prompt, without any weight updates. This is a far more flexible and cost-effective approach than fine-tuning. A frontier model like a hypothetical GPT-5 or Claude-4.5 could potentially become a powerful code-to-metric regressor on the fly, adapted to novel metrics or hardware simply by being shown a few examples. The paper misses the opportunity to investigate whether its proposed task can be solved "in-context," which is a crucial question for practical applicability.

- Dependence on Textual Representation: The model's performance is contingent on the quality and completeness of the textual representation of the code or computation graph. For complex structures, serializing them into a linear text format might lead to a loss of information.

**Questions:**

See weakness

---

> ### Author Response · Authors · 2025-11-15
>
> Hi Reviewer AHw1 - thanks for the feedback on our paper. Please find our responses below, including additional experiments as requested.
>
> > modern SOTA models excel at generalizing from just a handful of examples
>
> In-context learning (ICL) vs. weight adaptation: Numerous papers [1,2,3] have studied this general concept in detail, with the conclusion that both work well, with one beating the other depending on various conditions. Intuitively, assuming an arbitrarily long context window, ICL would be suitable, especially when bolstered by a huge service model with more knowledge.
> - But the fatal issue with ICL is its finite context window - with GPT-5’s 400K context window and fixed tokenization, only O(10) programs or NAS graphs can fit; this is far too few to be of serious use. Moreover, with such a limited memory state, it’s a very complex question for selecting the most informative examples to place and avoiding duplicate or similar examples over the input space.
> - Contrast this with in-weight adaptation, which in principle can freely absorb arbitrarily many examples (redundant or not) with sufficient model capacity, and whose memory doesn’t increase with the number of examples.
> - We performed a comparison against ICL over GPT5-Thinking (strongest model) [above](https://openreview.net/forum?id=KnzEMXSK3f&noteId=vNWKu8m6vP). As mentioned, due to the in-context token limit, we were only able to fit in at most 50 example pairs, and in all cases, ICL with GPT5-Thinking underperforms against the much smaller RLM.
>
> >  far more flexible and cost-effective approach than fine-tuning
>
> We respectfully disagree - relying on service LLMs has relatively huge inference costs, no control over pretraining knowledge, and no customization such as the vocabulary or tokenization. A user would be unable to control any part of the regression pipeline other than simple context. Furthermore, in-context learning is an emergent behavior that may be weaker with smaller service models (e.g. GPT Nano, Gemini Flash) as well.
>   - As shown in the table in our [general comments](https://openreview.net/forum?id=KnzEMXSK3f&noteId=vNWKu8m6vP), the RLM consistently outperforms in-context learning, as it can absorb significantly more examples in its weights. This becomes significantly more critical as one tries to build regressors for arbitrary formats not found in pretraining mixtures for service LLMs (e.g. ONNX graphs we presented). There are several use cases and domains that service LLMs have not been trained on, limiting their flexibility significantly.
>
> > regarding cost effectiveness
>
> The test in the table in [general comments](https://openreview.net/forum?id=KnzEMXSK3f&noteId=vNWKu8m6vP) for the MAX-Shot ICL costs 70M tokens when using a flagship LLM (GPT5, several billion parameter model), however, the RLM has permanently absorbed the task into its weights, and only costs ~0.83M tokens to evaluate, while being  OOM smaller (0.2-0.3B parameters). The true cost of the API calls associated with the MAX-Shot ICL test **far exceed even the total training cost of our largest RLM** (40 dollars to train the RLM on the cloud vs. 80 to only query the GPT5 model with MAX-Shot ICL on 256 examples). We also provide in  [general comments](https://openreview.net/forum?id=KnzEMXSK3f&noteId=vNWKu8m6vP) the exact token-costs for in-context learning, and find that RLMs at inference are roughly 45,000 times cheaper than ICL with flagship models.
>
> > Dependence on textual representation
>
> We disagree that textual representations are limiting; we believe they allow the ultimate flexibility for representing real world problems, especially code. One of the strongest examples of good performance with linear text format is our NAS results, which are extremely large, complex graph structures presented in ONNX format, containing all low-level execution details about the NAS architecture. Surprisingly, we find that the RLM matches SoTA graph neural networks at regressing on these complex structures. Although out-of-scope for this paper, we can even add the specifications of the hardware or read the low-level assembly language to have more fine-grained predictions too.
>
>
> [1] Chan, B., Chen, X., György, A., & Schuurmans, D. (2024). Toward understanding in-context vs. in-weight learning. arXiv preprint arXiv:2410.23042.
>
> [2] Lampinen, A. K., Chaudhry, A., Chan, S. C., Wild, C., Wan, D., Ku, A., ... & McClelland, J. L. (2025). On the generalization of language models from in-context learning and finetuning: a controlled study. arXiv preprint arXiv:2505.00661.
>
> [3] Mosbach, M., Pimentel, T., Ravfogel, S., Klakow, D., & Elazar, Y. (2023). Few-shot fine-tuning vs. in-context learning: A fair comparison and evaluation. arXiv preprint arXiv:2305.16938.

---

> > ### Author Response · Authors · 2025-11-25
> > **Kind follow-up on our rebuttal**
> >
> > Dear Reviewer AHw1,
> >
> > Hope you are doing well! We just wanted to briefly highlight that our rebuttal now:
> > - Adds direct comparisons between RLM and GPT-5-Thinking under 0/3/5/20/MAX-Shot ICL on Triton Kernels, APPS and NAS spaces, demonstrating that even MAX-shot ICL with a much larger flagship model under-performs the small RLM.
> > - Quantifies token and dollar costs of ICL vs. in-weight training, demonstrating that training and serving the RLM is substantially more cost effective than ICL with GPT-5-Thinking, while delivering significantly better ranking quality.
> >
> > We would be very grateful if you could take a look and let us know if there is anything else we should clarify.
> >
> > Regards,
> >
> > The Authors

---

### Author Response · Authors · 2025-11-15
**Regarding ICL with flagship models**

We thank all of the reviewers for their time spent on reading our paper in detail and the good questions raised. Below, we address a common question, which is the comparison against in-context learning (ICL).

We used the largest GPT5 variant (GPT-5 thinking) which allows a maximum 400K token context limit, and performed in-context regression by adding random examples (x,y) into the context as well as appropriate ICL-prompting. We scaled the number of examples up to the maximum token limit (~50 examples), which we term “MAX-Shot ICL”. Below is a comparison to the RLM, when performance is measured by Spearman rank:

| Method/Dataset | RLM   | **GPT5-Thinking**                                      |       |        |        |                      |
|----------------|-------|---------------------------------------------------------|-------|--------|--------|----------------------|
|                |       | 0-Shot | 3-Shot ICL | 5-Shot ICL | 20-Shot ICL | MAX-Shot ICL (~50) |
| Triton Kernels (Code)    | **0.562** | 0.230  | 0.216      | 0.432      | 0.465       | 0.456              |
| APPS (Code)    | **0.900** | -0.379 | 0.119      | 0.422      | 0.633       | 0.596              |
| DARTS (NAS)    | **0.672** | 0.074  | 0.310      | 0.430      | 0.470       | 0.544              |
| Amoeba (NAS)   | **0.751** | -0.017 | 0.263      | 0.340      | 0.344       | 0.413              |

As you can see, despite using a much larger and presumably more capable base model (GPT5), ICL performs consistently worse than in-weight training of the RLM due to the massive reduction in contextual examples used. Furthermore, the cost to train the much smaller and more customizable RLM (40 dollars) is less than the cost to just perform inference with GPT5 (80 dollars), demonstrating our method’s cost-effectiveness.

To further elaborate on the associated costs, please find below the average and maximum tokens required for the results above. As presented, for just one inference at MAX-Shot ICL, it can cost around $0.3125 per query (excluding thinking and output token costs). For comparison, the RLM can run on a single A6000 rented at roughly the same cost for an hour, serving ~12.5 queries per second (45 thousand queries at the same cost), all while significantly out-performing ICL with flagship models.


| Method/Dataset        | RLM (avg / max)  | 0-Shot ICL (avg / max) | 3-Shot ICL (avg / max)   | 5-Shot ICL (avg / max)    | 20-Shot ICL (avg / max)    | MAX-Shot ICL (~50) (avg / max) |
|-----------------------|------------------|------------------------|--------------------------|---------------------------|----------------------------|---------------------------------|
| Triton Kernels (Code) | 3223.5 / 8251    | 3223.5 / 8251          | 13986.8 / 30112          | 52787.0 / 141911          | 72719.5 / 102817           | ~272000 / ~272000              |
| APPS    (Code)              | 1338.0 / 1521    | 1338.0 / 1521          | 4998.5 / 5255            | 7734.8 / 8876             | 26785.8 / 29854            | ~272000 / ~272000              |
| DARTS (NAS)           | 4234.0 / 4234    | 4234.0 / 4234          | 16612.0 / 16612          | 24864.0 / 24864           | 86764.5 / 86765            | ~272000 / ~272000              |
| Amoeba (NAS)          | 4234.0 / 4234    | 4234.0 / 4234          | 16611.8 / 16612          | 24863.8 / 24864           | 86764.5 / 86765            | ~272000 / ~272000              |

---

### Author Response · Authors · 2025-12-03
**Summary of reviewer-discussions**

# General Response

We would like to than the reviewers for their positive feedback and thorough review. To help the AC quickly understand where the paper stands after the discussion, we summarize our core contributions and how the reviewer concerns were addressed in the rebuttal.

Our paper studies code-to-metric regression at scale, predicting numerical properties such as memory, latency and model accuracy from code or ONNX text, without _any hand designed features or static analyzers_.

We show that a single ~300M parameter RLM initialized from T5Gemma, equipped with a numerical tokenizer can:
- Predict memory usage for competitive programming submissions on APPS with Spearman ~ 0.9.
- Achieve non-trivial rank correlation across 17 CodeNet languages, with a single unified model.
- Predict Triton GPU kernel latency from kernel text alone.
- Match or slightly exceed state-of-the-art NAS regressors (e.g., FLAN variants) on five classic NAS spaces, using only ONNX text (no graph kernels, no zero-cost proxies), while also predicting latencies on four heterogeneous hardware platforms in a multi-objective setting.

We added substantial experiments to target the main questions:
- Strong ICL and large-model baselines: we ran 0/3/5/20/MAX-Shot ICL with GPT-5-Thinking (400K context) on APPS, Triton and NAS spaces. Even when maxing out the context length with in-context examples, GPT-5 underperforms a small RLM, while being orders of magnitude more expensive per query. For example, MAX-Shot ICL ablation uses 272K tokens per query, and costs more to run than the cost of training the RLM end to end! Under the same budget, the RLM can serve 45,000 times more queries than GPT-5-Max-Shot ICL while performing better.

- Alternative backbones (Qwen, Llama): At request of Reviewer sjYR, we trained RLMs on Qwen-3-0.6B and Llama-3.2-1B using their default vocab, both severely underform T5Gemma based RLM, supporting our encoder-based design choice as well as custom numerical decoder.

- We ran an ablation comparing a plain, log-transformed and normalized regression head, all performing worse than the decoding head we study.

- Demonstrated that APPS has significant label variance.

- Provided CRPS and NLL for multi-objective settings, showing well-calibrated predictive distributions.

In summary, we show that a small, specialized RLM can reliably regress memory, latency and accuracy directly from code and ONNX representations of graphs, matching or surpassing strong baselines, while being several orders of magnitude cheaper than large model-ICL based approaches. Our ablations on architecture, decoder choice, and multi-task + multi-objective training take a first major step towards practical, feature-engineering free code-to-metric regression.

---

### Meta-Review · Area_Chair_MMU9 · 2025-12-15

**Summary:**

This paper introduces RLM, encoder-decoder models trained to predict performance metrics given code snippets. RLMs do so by predicting textual representation of such metrics, so that training is carried out via standard conditional maximum likelihood estimation. Authors evaluate RLMs instantiated from pre-trained and general purpose encoder-decoder models, and show that, upon fine-tune, somewhat small models are able to reach relatively high correlation with group truth performance.

**Reviewer Concerns:**

Reviewers concerns mostly revolved around the strength of the empirical evidence presented in the paper. Main concerns raised by reviewers in my opinion involved lack of stronger baselines, to which authors replied with extra results based on ICL evaluations of GPT-5, and experiments designed to evaluate the extent to which RLM actually rely on simple patterns in the data (e.g., if number of LOC correlates with latency, a model would easily pick on that spurious feature). I do tend to agree with the reviewer's concerns and would say a deeper evaluation is required to try and understand failure cases and possible spurious biases models might be exploiting. In code retrieval settings for instance, it's well known now that models exploit shallow lexical matching and break upon randomisation of entity names. While results here seem promising, a deeper empirical assessment would be require to support the main claims in the paper.

**Reviewer Scores:**

e7RV, rating: 4. Would likely maintain the original score as main concerns involved expanding the evaluation to larger models and different target representations, which authors replied to partially.
AHw1, rating: 4. Would also likely maintain. Concerns revolved around lack of novelty, based on the observation that the text-to-text task is known to be solvable by encoder-decoder models in many many setting, so most tasks posed in that way are expected to be solved by this model class given enough data and compute. The reviewer also raised concerns regarding the evaluation metrics, to which authors replied satisfactorily in my opinion.
NB5z, rating: 6. The reviews very low confidence and would thus likely not change.
sjYR, rating: 4. The reviewer's main concern was related to baselines, and the reviewer seemed satisfied with the added experiments and would very likely have bumped their score up.

---

### Decision · Program_Chairs · 2026-01-26

Reject